# CHOICE: Benchmarking the Remote Sensing Capabilities of Large Vision-Language Models

**Xiao An**\*, **Jiaxing Sun**\*, **Zihan Gui**, **Wei He**†
State Key Lab. LIESMARS, Wuhan University
{anxiao, sunjiax0323, syro_gzh, weihe1990}@whu.edu.cn

## Abstract

The rapid advancement of Large Vision-Language Models (VLMs), both general-domain models and those specifically tailored for remote sensing, has demonstrated exceptional perception and reasoning capabilities in Earth observation tasks. However, a benchmark for systematically evaluating their capabilities in this domain is still lacking. To bridge this gap, we propose CHOICE, an extensive benchmark designed to objectively evaluate the hierarchical remote sensing capabilities of VLMs. Focusing on 2 primary capability dimensions essential to remote sensing: perception and reasoning, we further categorize 6 secondary dimensions and 23 leaf tasks to ensure a well-rounded assessment coverage. CHOICE guarantees the quality of all 10,507 problems through a rigorous process of data collection from 50 globally distributed cities, question construction, and quality control. The newly curated data and the format of multiple-choice questions with definitive answers allow for an objective and straightforward performance assessment. Our evaluation of 3 proprietary and 21 open-source VLMs highlights their critical limitations within this specialized context. We hope that CHOICE will serve as a valuable resource and offer deeper insights into the challenges and potential of VLMs in the field of remote sensing. Code and dataset are available at this https URL.

## 1 Introduction

In recent years, Large Language Models (LLMs) [1, 2] have gained significant traction in natural language processing, emerging as powerful tools for understanding vast amounts of text data, reasoning over complex language patterns, and generating coherent and contextually relevant responses as a know-it-all. Building upon the success of LLMs, general-domain Large Vision-Language Models (VLMs) like Qwen2-VL [3], Llama3.2 [4] and InternVL2 [5] have also reformed the conventional modes of visual interaction, exhibiting impressive capabilities in comprehending visual-linguistic information, reasoning, and responding accurately to intricate human queries. Remote sensing, a field dedicated to observing the physical world from an aerial perspective [6, 7], distinct from the ground view, has also experienced revolutionary advancements thanks to the emerging Remote Sensing Large Vision-Language Models (RSVLMs), such as RemoteCLIP [8], GeoChat [9], LHRS-Bot [10]. Benefiting from their specialized perceiving and reasoning capabilities for Remote Sensing Images (RSIs), RSVLMs facilitate multiple downstream tasks within a unified framework, including classification, visual grounding, and visual question answering.

A systematic and objective evaluation of model capabilities not only reflects the progress of VLMs but also guides future advancements. The thriving research communities have introduced several benchmarks, such as MM-Bench [11], SEED-Bench [12], and MMStar [13], for assessing the versatile capabilities of general-domain VLMs in the context of the natural, everyday world. However,

---

\*Equal contribution. †Corresponding author.

39th Conference on Neural Information Processing Systems (NeurIPS 2025) Track on Datasets and Benchmarks.

Table 1: Comparison of CHOICE with existing datasets & benchmarks. "All-new" shows whether the dataset or benchmark excludes publicly available datasets and is not involved in the training of VLMs. "Dimension" means the number of fine-grained evaluation capabilities. Abbreviations adopted: O for Optical; MI for Multi-Image; V for Video; BT for Bi-Temporal; MT for Multi-Temporal; MCQ for Multi-Choice Question; FF for Free Form; BBox for Bounding Box; Seg for Segmentation Mask.

| Benchmark/Dataset | Domain | Modalities | Data Sources | All-new | Geospatial Coverage | Answer Type | Reasoning | Dimensions |
|---|---|---|---|---|---|---|---|---|
| MMBench | General | O, MI | Multiple Public Datasets | ✗ | N/A | MCQ | ✓ | 20 |
| MMStar | General | O, MI | Multiple Public Benchmarks | ✗ | N/A | MCQ | ✓ | 18 |
| SEED-Bench-2 | General | O, MI, V | Multiple Public Datasets | ✗ | N/A | MCQ | ✓ | 12 |
| UCM-Caption | RS | O | UCM | ✗ | US Regions | FF | ✗ | 1 |
| RemoteCount | RS | O | DOTA | ✗ | Multiple Cities | FF | ✗ | 1 |
| LEVIR-CC | RS | O, BT | LEVIR-CD | ✗ | 20 Regions in Texas | FF | ✗ | 1 |
| DIOR-RSVG | RS | O | DIOR | ✗ | >80 Countries | BBox | ✗ | 1 |
| RSVQA-LR/HR | RS | O | Sentinel-2, USGS | ✓ | Netherlands & USA | FF | ✓ | 1 |
| RSIEval | RS | O | DOTA | ✗ | Multiple Cities | FF | ✓ | 6 |
| LHRS-Bench | RS | O | Google Earth+OSM | ✓ | N/A | MCQ | ✓ | 11 |
| GeoChat-Bench | RS | O | SAMRS | ✗ | Multiple Regions | FF, BBox | ✗ | 6 |
| EarthVQA | RS | O | LoveDA, WorldView-3 | ✗ | 18 Regions in 3 Cities | FF | ✗ | 6 |
| FineGrip | RS | O | MAR20 | ✗ | N/A | FF, Seg | ✗ | 5 |
| VRSBench | RS | O | DOTA, DIOR | ✗ | Multiple Regions | FF, BBox | ✓ | 3 |
| CHOICE | RS | O, MI, BT, MT | Diverse Platforms | ✓ | 50 Cities Worldwide | MCQ, BBox, Seg | ✓ | 23 |

due to **the domain gap between natural images and RSIs, as well as significant regional intra-class variations** [14], these results fail to directly reflect proficiency in remote sensing. Therefore, additional evaluation dimensions beyond standard paradigms are required to assess aerial-specific capabilities effectively.

As shown in Table 1, current evaluations of VLMs in remote sensing are subjected to several limitations: **(1) Separate scope:** the prevailing evaluation has long relied on a handful of individual datasets, each targeting only one specific skill. For instance, UCM-Caption [15] is used for image captioning, LEVIR-CC [16] for change detection, and DIOR-RSVG [17] for visual grounding. However, the lack of local information analysis and multi-temporal reasoning of RSIs, which are key aspects in remote sensing, and disparities in data formats, quality, and metrics across these datasets complicate the comprehensive evaluation. Moreover, current metrics require an exact match between model response and ground truth, leading to a rigid and biased limitation. **(2) Fragmented benchmarking:** recent studies have proposed new benchmarks, including LHRS-Bench [10], EarthVQA [18] and VRSBench [19], for more specialized remote sensing tasks. However, the coarse taxonomy of evaluation dimensions and the limited quantity of samples and tasks (especially the absence of pixel-wise and multi-temporal tasks) offer only a fragmented perspective on in-depth hierarchical capabilities, lagging behind the rapid development of VLMs. **(3) Non-objectivity by Data leakage:** paramountly, most mainstream datasets or benchmarks in remote sensing are repurposed from common datasets, some of which are engaged in the training stages of VLMs, such as DOTA [20], DIOR [21]. This recycling of data leads to inevitable data leakage and undermines objectivity during the evaluation process (See more details of data leakage in Appendix B).

To fill this gap, we propose CHOICE, a multi-modal benchmark specifically designed to systematically and objectively assess the remote sensing capabilities of VLMs. As illustrated in Figure 1, our taxonomy categorizes perception and reasoning as Level-1 (L-1) capabilities, which are further subdivided into 6 L-2 dimensions and 23 L-3 leaf tasks for a more in-depth evaluation. **CHOICE is the first benchmark that provides both hierarchical and objective evaluation of the remote sensing capabilities of VLMs**. Specifically, to reduce the bias inherent in existing metrics, CHOICE consists of 9,407 multiple-choice questions (MCQs) spanning 21 of the 23 tasks, 600 visual grounding problems, and 500 reference expression segmentation cases, covering the majority of crucial downstream tasks in remote sensing. To avoid data leakage and ensure objectivity, we collect all RSIs worldwide from various satellites, platforms, and products on our own, intentionally excluding any publicly available datasets. Three methods are employed to construct the problems: (1) Label-driven Construction, (2) Foundation Model-driven Construction, and (3) Human-GPT4 Collaboration. Human annotators are further involved to ensure the quality and correctness. Leveraging the instruction-following ability of VLMs, we restrict responses to A/B/C/D options for MCQs, which simplifies the computation of accuracy and offers objective metrics for evaluation.

Based on CHOICE, the first multi-modal remote sensing benchmark with broad geospatial coverage across 50 globally distributed cities and diverse landforms, we conducted a hierarchical evaluation of

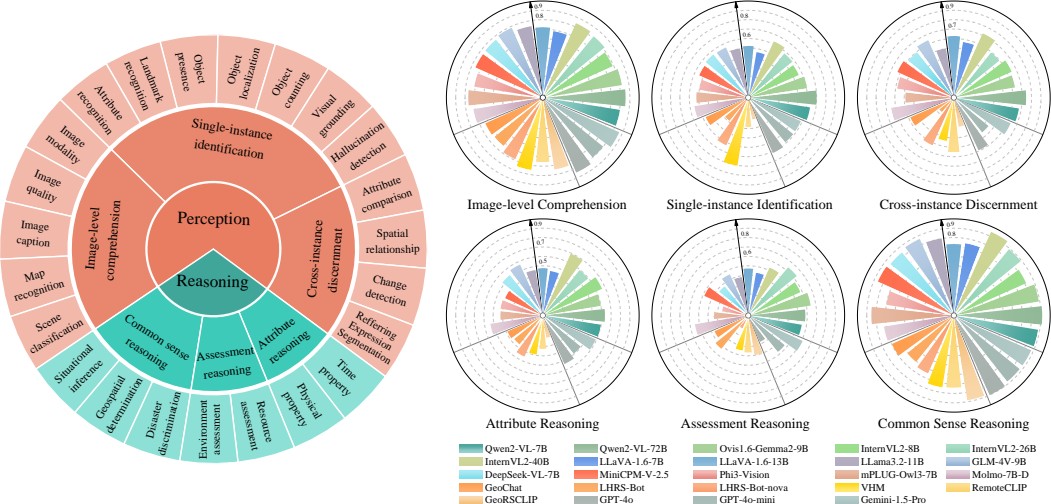

Figure 1: (left) The hierarchical capability taxonomy of CHOICE, which concentrates on perception and reasoning capabilities and is categorized into 6 Level-2 dimensions and 23 Level-3 leaf tasks. (right) Evaluation results of the 6 Level-2 capability dimensions across mainstream VLMs. Each circle is divided into three parts by three gray lines, which, ordered by area from largest to smallest, correspond to general-domain open-source VLMs, RSVLMs, and proprietary VLMs, respectively.

proprietary VLMs, mainstream general-domain VLMs, RSVLMs, and CLIP-based VLMs in remote sensing. The experimental results yield the following three key findings: **(1) RSVLMs excel in tasks that require a high degree of specialized remote sensing knowledge, while showing no clear consistent superiority over general-domain VLMs:** the domain gap underlines the necessity of domain-specific data for optimal performance. However, the neglect of integration with general knowledge contributes to the underperformance of RSVLMs. **(2) Fine-grained perception and reasoning are key challenges for VLMs:** the perception of fine-grained objects and advanced reasoning tasks involving complex scenes, social attributes and specific remote sensing characteristics pose significant challenges for nearly all evaluated VLMs; **(3) Open-source VLMs can serve as viable alternatives to proprietary VLMs in remote sensing:** cutting-edge VLMs like Qwen2-VL-70B and InternVL2-40B demonstrate competitive or even superior performance in specific tasks compared to GPT-4o, showcasing the great potential of open-source VLMs in advancing remote sensing applications.

## 2 Related Works

**Large Vision-Language Models.** Prior CLIP-based studies have demonstrated remarkable zero-shot performance with natural language supervision [22]. Building upon the success of Large Language Models (LLMs) [1, 2], recent advancements in large vision-language models (VLM) have also made significant strides as well [3–5]. By aligning visual-linguistic representations with image-text pairs and instruction-finetuning, these VLMs achieve a holistic and precise understanding of the natural world. For more specialized applications, several remote sensing large vision-language models (RSVLMs) have been proposed to interpret complex RSIs. While CLIP-based models are constrained to coarse image-level interpretation [8, 23], generative RSVLMs are well-equipped to handle more complex tasks. GeoChat [9] excels in visual grounding tasks, while LHRS-Bot [10] supports more fine-grained perception tasks like instance attributes and relationships. Additionally, VHM [24] advances visual grounding and explores model hallucination mitigation. Though qualitative results in remote sensing have been encouraging, a systematic quantitative evaluation is of great necessity to evaluate and compare the remote sensing capabilities of different VLMs.

**Benchmarks for Large Vision-Language Models.** Several concurrent works [11, 12, 25] have proposed versatile benchmarks to evaluate VLMs from the perspective of the natural, everyday world. Within the realm of remote sensing capabilities, recent works [8, 26] have evaluated their models using existing public multimodal datasets, which are usually further processed from common datasets

- some of which are engaged during the pretraining stage of VLMs, such as UCM-Captions [15], RSVQA [27], LEVIR-CC [16], DIOR-RSVG [17]. This method introduces inevitable non-objectivity to the evaluation procedure. Moreover, these datasets are designed only for one or a few specific evaluation dimensions, rendering them inadequate for a systematic evaluation of the remote sensing capabilities of VLMs. While some benchmarks are released alongside their VLMs [9, 10, 19], the coarse evaluation dimensions and limited quantity of problems fail to keep up with the advancements in VLMs.

## 3 CHOICE

### 3.1 Hierarchical Capability Taxonomy

As shown in Figure 1, we have meticulously structured the evaluation hierarchies of CHOICE into three levels, which comprise a significantly broader range of evaluation dimensions than the combined offerings of all existing datasets listed in Table 1. Perception and Reasoning are the 2 L-1 capabilities, which reflect the extent to which VLMs can extract information from RSI inputs and the conclusions they can draw from this information, respectively. L-1 capabilities are divided into 6 fine-grained L-2 dimensions, further resulting in a total of 23 L-3 tasks tailored to remote sensing. Detailed definitions are illustrated below and Appendix A.

#### 3.1.1 Perception

Remote sensing is dedicated to perceiving the physical world from a bird's-eye view, with a single RSVLM now capable of accomplishing most perception tasks [28, 29]. Given the multi-satellite sources and the domain gap between natural images and RSIs, additional evaluation dimensions beyond the standard paradigms [11, 12] are required to assess aerial-specific perception capabilities. Remote sensing emphasizes three perception tiers: image-wise, instance-wise, and pixel-wise. Therefore, we categorized the perception capability into three components: 1) image-level comprehension; 2) single-instance identification, and 3) cross-instance discernment (including pixel-wise task). Figure 2 illustrates some examples of Perception in CHOICE, and the detailed information for each category is provided below.

**Image-level Comprehension (ILC)** provides a coarse yet holistic interpretation of RSIs, capturing overall semantics and field-specific attributes [30], which enables insights into the image's content and thematic relevance. This tier consists of five leaf tasks: Scene Classification (SC), Map Recognition (MR), Image Caption (IC), Image Quality (IQ), and Image Modality (IM).

**Single-instance Identification (SII)** is a fine-grained task in remote sensing, aiming to precisely identify and locate specific objects within an image [31]. It enables detailed object recognition and localization in an open-vocabulary context, supporting dynamic, context-sensitive analysis without relying on fixed categories. This tier comprises seven tasks: Landmark Recognition (LR), Object Presence (OP), Object Localization (OL), Object Counting (OC), Attribute Recognition (AR), Visual Grounding (VG), and Hallucination Detection (HD).

**Cross-instance Discernment (CID)** extends SII by emphasizing comparisons or changes among multiple objects [16]. This advanced dimension focuses on analyzing relationships, interactions, and transformations, providing insights into contextual dependencies, spatial dynamics, and temporal changes within images. This tier contains four leaf tasks: Attribute Comparison (AC), Spatial Relationship (SR), Change Detection (CD), and Referring Expression Segmentation (RES).

#### 3.1.2 Reasoning

Once context information is perceived, determining the extent to which conclusions can be drawn based on visual inputs and common sense embedded in LLMs becomes another essential capability for VLMs: reasoning. Remote sensing mainly focuses on reasoning scene or instance attributes, evaluating environmental context, and deducing geospatial information. While existing datasets concentrate on perception and there is a noticeable gap when it comes to assessing remote sensing reasoning capabilities within the current VLM benchmarks, our CHOICE fills this void by establishing three L-2 dimensions, meticulously designed to gauge proficiency in field-specific reasoning tasks pertinent to remote sensing applications. Figure 2 presents some examples of Reasoning in CHOICE, and the detailed information for each category is as follows:

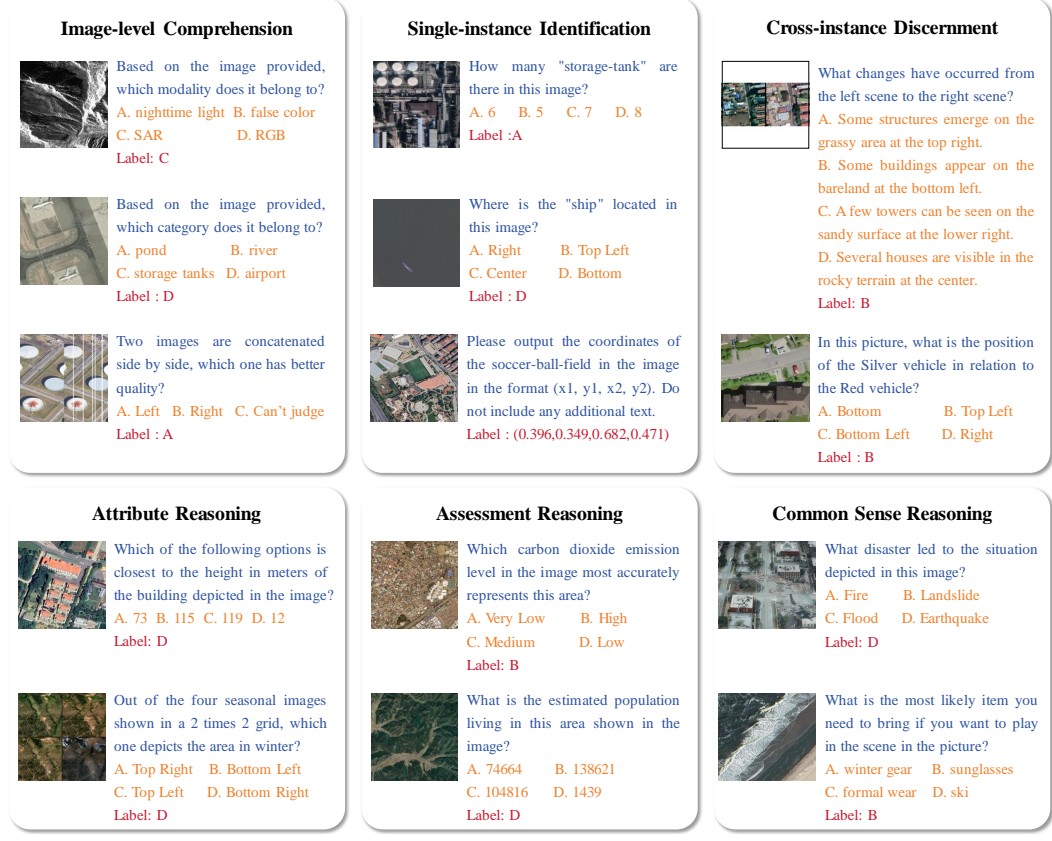

Figure 2: Examples of CHOICE from the 6 L-2 capabilities.

**Attribute Reasoning (AttR)** includes two key aspects: Time Property (TP) and Physical Property (PP). This dimension enables reasoning about temporal information specific to RSIs, like seasonal images, and assesses physical attributes requiring real-world measurement [32], providing insights into both chronological context and tangible characteristics within the imagery.

**Assessment Reasoning (AssR)** is an advanced VLM capability that enables estimation of societal indicators, like population density [33], and assessment of environmental conditions depicted in RSIs, such as monthly $CO_2$ emissions [34]. This dimension allows VLMs to derive high-level insights regarding both human and environmental factors from RSIs, facilitating informed and effective decision-making. Resource Assessment (RA) and Environmental Assessment (EA) are included in this tier.

**Common Sense Reasoning (CSR)** is also a crucial dimension to evaluate the performance of VLMs, which are built upon the LLMs embedded with extensive common sense knowledge. Applying CSR to remote sensing, however, remains largely unexplored, requiring models to infer everyday knowledge and contextual understanding directly from RSIs. To address this gap, we introduce three L-2 dimensions within this tier: Disaster Discrimination (DD), Geospatial Determination (GD), and Situation Inference (SI).

## 3.2 Construction of CHOICE

CHOICE encompasses 23 remote sensing-specific leaf tasks, culminating in a total of 10,507 problems detailed in Appendix A.2. Except for VG and RES, all problems in the other tasks are formatted as MCQs $P_i = [Q_i, C_i, I_i, L_i]$, where $Q_i$ denotes the question, $C_i$ represents a set of $n(2 \leq n \leq 4)$ choices $c_1, c_2, \ldots, c_n$, $I_i$ is the associated RSI for the problem $P_i$, and $L_i$ is the correct label. For VG and RES, we omit $C_i$ and instead assign $L_i$ as the coordinates and segmentation mask of the object mentioned in $Q_i$, respectively. All the RSIs are collected manually from multi-source satellites, without any inclusion of publicly available datasets, which ensures the objectivity of evaluation.

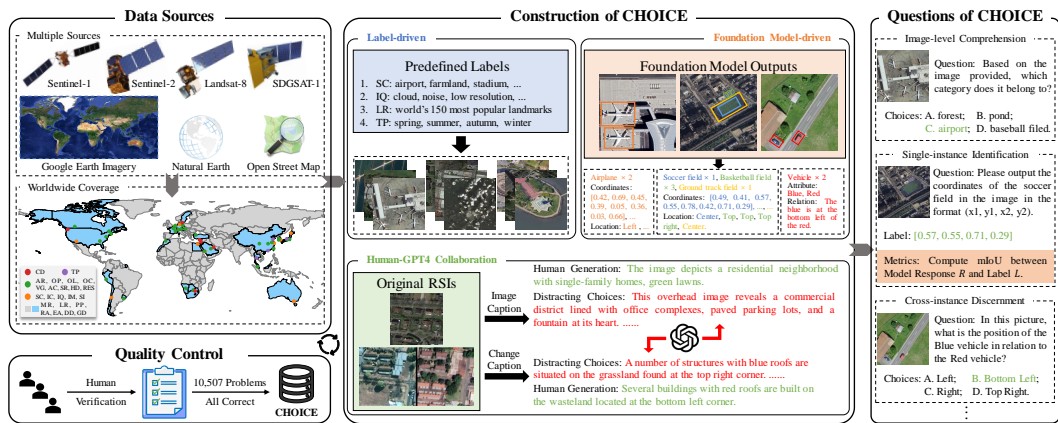

Figure 3: Overview of the construction of CHOICE. All RSIs, sourced around the world, are collected from various satellites, products, and platforms. There are three approaches to generating the problems and a human-involved quality control process guarantees the accuracy of all the questions.

Additionally, all the questions and choices are designed by hand or with the assistance of GPT-4 [35] based on the predefined capabilities and tasks. The construction of CHOICE is detailed as follows and Figure 3.

**Data Coverage Area.** RSIs exhibit significant regional intra-class variations, particularly from a global viewpoint. However, existing evaluation datasets are limited in geographic coverage [36, 16, 17]. To enhance the data diversity, the majority of RSIs in CHOICE originate from 50 randomly chosen cities (including both urban and rural areas) spanning all six continents (excluding Antarctica), based on the largest 1,000 cities across 163 countries identified by Oxford Economics' GCI [37], which eliminates the potential biases within the constructed problems. Specifically, for tasks with predetermined locations, or where $I_i$ is required to cover a large area or has low spatial resolution, we extend the coverage globally and randomly select areas to ensure substantial original data.

**Benchmark Statistics.** We constructed 10,507 problems in total across 23 L-3 tasks, with RSIs sourced from multiple satellites, products, and platforms, including Landsat-8, Sentinel-1/2, Google Earth Engine, etc. The spatial resolution varies from 0.1 meter/pixel to 30 meter/pixel, with high-detail tasks focusing on instances, e.g., Object Counting and Visual Grounding, having a resolution of 0.1 meter/pixel to 0.3 meter/pixel. The image sizes range from $512 \times 512$ to $800 \times 800$. To ensure a balanced evaluation, we aimed for an even distribution among problems associated with different capabilities during data collection, with a minimum of 150 samples per task. Further details are provided in Appendix A.2.

**Construction of MCQs.** We employ three approaches to collect RSIs and create associated questions: (1) collect RSIs based on the predefined labels and pick distracting choices from the labels of other samples; (2) utilize various conventional foundation models to extract relevant visual information from the huge volume of RSIs and generate distracting choices according to this information; (3) human creation combined with GPT-4. Details are shown below and in Figure 3.

*Label-driven Construction.* For evaluation dimensions like Image-level Comprehension and Attribute Reasoning, we predefine a set of labels, such as categories for scene classification task and seasonal tags according to the time the RSIs were captured for time property task. With the assistance of OpenStreetMap (OSM) [38], a crowdsourced volunteered geographic information database containing abundant geographical objects, we manually collect RSIs from Google Earth Engine [39], where RSIs containing the locations of interest or captured on designated dates are available, then assign the label as the correct answer (See Figure 11 for an example). Distracting choices are crafted from labels of other samples for the corresponding problems.

*Foundation Model-driven Construction.* For evaluation dimensions requiring detailed instance-wise information, various visual foundation models [40–42] are incorporated to accurately capture object attributes, particularly the coordinates of rotated bounding box for Visual Grounding, which serve as the foundation for related tasks. We take the intersection of outputs predicted with at least

90% accuracy and then employ human annotators to verify all the attributes, such as bounding box coordinates, location, direction, color (obtained by K-Means clustering), and segmentation masks. Notably, we adopt rotated bounding boxes as labels for Visual Grounding due to the varied orientations of objects in RSIs. We hope more VLMs will support outputting coordinates in this format. Finally, we manually construct the problems based on the verified information for fine-grained instance-wise tasks. See details in Appendix C.

*Human-GPT4 Collaboration.* Tasks that emphasize changes between scenes and make inferences based on RSIs are challenging because no foundation models can yet describe these complex relations in natural language accurately. Therefore, we enlist human annotators to create precise descriptions, including captions for RSIs and explanations of differences between paired RSIs captured at different timestamps. Additionally, we generate three distractors using GPT-4 [35], followed by human verification to filter out options that are too similar to the ground truth.

**Quality Control.** We implemented a multi-stage quality control process by recruiting 12 volunteers with professional backgrounds in remote sensing or computer vision, including both master's and doctoral students, to assist in the construction and quality assurance of CHOICE. For detailed procedures and further information, please refer to Appendix F.

## 3.3 Evaluation Strategy

**Evaluation of LLM-based VLMs.** Leveraging the extraordinary instruction-following abilities, VLMs can directly output the choice "A" or "B" or "C" or "D", or the content of the choice, which facilitates subsequent choice extraction and accuracy calculation. Specifically for Visual Grounding, we extract the coordinates output by VLMs, and accuracy is calculated if the predicted box has an overlap of more than 0.5 IoU with the ground-truth box. The correct answer of MCQs is randomly assigned among the choices, eliminating any bias from answer ordering.

**Evaluation of CLIP-based VLMs.** Since CLIP-based VLMs lack language output abilities, we adopt the method in [8] to augment each question and $n$ associated choices to $n$ declarative sentences, and calculate the similarity score between the RSI and each sentence. The choice in the sentence that achieves the highest similarity score with the RSI is considered the output of VLMs.

# 4 Evaluation Results

## 4.1 Experimental Setup

We evaluated 24 mainstream VLMs based on CHOICE, which can be categorized into three groups: 15 open-source general-domain VLMs, including Qwen2-VL series [3], Ovis1.6-Gemma2-9B [43], InternVL2 series [5], LLaVA-1.6 series [44], Llama3.2 [4], GLM-4V [45], DeepSeek-VL [46], MiniCPM-V [47], Phi3-Vision [48], mPLUG-Owl3 [49] and Molmo [50], 6 RSVLMs: GeoChat [9], LHRS-Bot [10], LHRS-Bot-nova [51], VHM [24], RemoteCLIP [8], and GeoRSCLIP [23], and 3 proprietary VLMs: GPT-4o-mini, GPT-4o-2024-11-20 [52], and Gemini-1.5-Pro [53]. All VLMs in our evaluation follow the same zero-shot prompt strategy using their default generation configurations. Since most of our problems are MCQs, this format ensures the objectivity and rigor of the evaluation. For Visual Grounding task, we set the IoU metric (>0.5 is considered correct) to compute the accuracy.

## 4.2 Main Results

The overall evaluation results of CHOICE are depicted in Figure 1 and Table 2. Currently, general-domain VLMs, benefiting from extensive training corpora, exhibit a broader range of capabilities, even outperforming RSVLMs in certain abilities pertinent to remote sensing. However, none of the VLMs achieve consistently high performance across the 6 capability dimensions of CHOICE, highlighting substantial room for improvement in tackling complex remote sensing tasks.

**ILC and CSR Dimensions.** The results for these two L-2 dimensions indicate that the majority of general-domain VLMs and RSVLMs exhibit strong capabilities in ILC and CSR, with task accuracy rates reaching approximately up to 80% or higher. This outcome demonstrates that these VLMs generally possess the foundational and critical capabilities for reasoning about common sense knowledge in remote sensing scenarios and comprehending RSIs at the image level.

Table 2: Evaluation results for L-2 dimensions. ILC for Image-level Comprehension; SII for Single-instance Identification; CID for Cross-instance Discernment; AttR for Attribute Reasoning; AssR for Assessment Reasoning; CSR for common sense Reasoning. The best (second best) is in bold (underline).

| Model | ILC | SII | CID | AttR | AssR | CSR |
|---|---|---|---|---|---|---|
| **Proprietary Large Vision-Language Models** | | | | | | |
| GPT-4o-mini | 0.800 | 0.588 | 0.448 | 0.494 | 0.474 | 0.876 |
| GPT-4o-2024-11-20 | 0.845 | 0.616 | 0.591 | 0.536 | 0.277 | 0.900 |
| Gemini-1.5-Pro | **0.867** | 0.585 | 0.636 | 0.590 | 0.611 | 0.876 |
| **General-domain Large Vision-Language Models** | | | | | | |
| Qwen2-VL-7B | 0.806 | 0.638 | 0.675 | 0.600 | 0.550 | 0.884 |
| Qwen2-VL-72B | 0.855 | 0.702 | **0.742** | 0.634 | 0.580 | 0.914 |
| Ovis1.6-Gemma2-9B | 0.828 | 0.606 | 0.632 | 0.598 | **0.645** | 0.890 |
| InternVL2-8B | 0.789 | 0.566 | 0.648 | 0.664 | 0.580 | 0.816 |
| InternVL2-26B | 0.820 | 0.578 | 0.595 | 0.594 | 0.638 | 0.890 |
| InternVL2-40B | 0.838 | 0.629 | 0.721 | **0.694** | 0.530 | **0.946** |
| LLaVA-1.6-7B | 0.686 | 0.463 | 0.574 | 0.450 | 0.433 | 0.749 |
| LLaVA-1.6-13B | 0.719 | 0.522 | 0.626 | 0.474 | 0.470 | 0.733 |
| Llama3.2-11B | 0.746 | 0.502 | 0.504 | 0.460 | 0.388 | 0.803 |
| GLM-4V-9B | 0.785 | 0.555 | 0.644 | 0.564 | 0.450 | 0.863 |
| DeepSeek-VL-7B | 0.764 | 0.561 | 0.595 | 0.528 | 0.373 | 0.840 |
| MiniCPM-V-2.5 | 0.757 | 0.546 | 0.631 | 0.414 | 0.493 | 0.857 |
| Phi3-Vision | 0.710 | 0.503 | 0.582 | 0.428 | 0.253 | 0.700 |
| mPLUG-Owl3-7B | 0.766 | 0.524 | 0.487 | 0.422 | 0.340 | 0.850 |
| Molmo-7B-D | 0.720 | 0.553 | 0.648 | 0.536 | 0.548 | 0.724 |
| **Remote Sensing Large Vision-Language Models** | | | | | | |
| GeoChat | 0.642 | 0.469 | 0.480 | 0.384 | 0.368 | 0.697 |
| LHRS-Bot | 0.633 | 0.290 | 0.171 | 0.366 | 0.426 | 0.610 |
| LHRS-Bot-nova | 0.688 | 0.530 | 0.526 | 0.450 | 0.120 | 0.644 |
| VHM | 0.751 | **0.703** | 0.436 | 0.392 | 0.348 | 0.744 |
| RemoteCLIP | 0.657 | 0.283 | 0.552 | 0.326 | 0.364 | 0.739 |
| GeoRSCLIP | 0.745 | 0.198 | 0.285 | 0.210 | 0.397 | 0.884 |

Table 3: Fine-grained evaluation results for Reasoning. TP for Time Property; PP for Physical Property; EA for Environmental Assessment; RA for Resource Assessment; DD for Disaster Discrimination; GD for Geospatial Determination; SI for Situation Inference. The best (second best) is in bold (underline).

| Model | AttR | | AssR | | | CSR | SI |
|---|---|---|---|---|---|---|---|
| | TP | PP | EA | RA | DD | GD | |
| **Proprietary Large Vision-Language Models** | | | | | | | |
| GPT-4o-mini | 0.420 | 0.543 | 0.420 | 0.492 | 0.870 | 0.925 | 0.847 |
| GPT-4o-2024-11-20 | 0.565 | 0.517 | 0.470 | 0.213 | 0.890 | 0.950 | 0.873 |
| Gemini-1.5-Pro | 0.520 | 0.637 | 0.460 | **0.662** | 0.835 | 0.985 | 0.830 |
| **General-domain Large Vision-Language Models** | | | | | | | |
| Qwen2-VL-7B | 0.510 | 0.660 | 0.580 | 0.540 | 0.880 | 0.970 | 0.830 |
| Qwen2-VL-72B | 0.580 | 0.670 | 0.490 | 0.610 | 0.900 | 0.980 | 0.880 |
| Ovis1.6-Gemma2-9B | 0.580 | 0.610 | 0.600 | 0.660 | 0.910 | 0.930 | 0.850 |
| InternVL2-8B | 0.520 | **0.760** | 0.400 | 0.640 | 0.880 | 0.790 | 0.790 |
| InternVL2-26B | 0.510 | 0.650 | **0.690** | 0.620 | 0.850 | 0.960 | 0.870 |
| InternVL2-40B | **0.610** | 0.750 | 0.560 | 0.520 | **0.950** | 0.980 | **0.920** |
| LLaVA-1.6-7B | 0.300 | 0.550 | 0.320 | 0.470 | 0.740 | 0.710 | 0.780 |
| LLaVA-1.6-13B | 0.330 | 0.570 | 0.560 | 0.440 | 0.740 | 0.640 | 0.790 |
| Llama3.2-11B | 0.430 | 0.480 | 0.440 | 0.370 | 0.810 | 0.890 | 0.740 |
| GLM-4V-9B | 0.420 | 0.660 | 0.360 | 0.480 | 0.860 | **0.990** | 0.780 |
| DeepSeek-VL-7B | 0.390 | 0.620 | 0.170 | 0.440 | 0.860 | 0.880 | 0.800 |
| MiniCPM-V-2.5 | 0.390 | 0.430 | 0.620 | 0.450 | 0.920 | 0.880 | 0.800 |
| Phi3-Vision | 0.380 | 0.460 | 0.290 | 0.240 | 0.850 | 0.580 | 0.680 |
| mPLUG-Owl3-7B | 0.350 | 0.470 | 0.520 | 0.280 | 0.900 | 0.860 | 0.810 |
| Molmo-7B-D | 0.500 | 0.560 | 0.600 | 0.530 | 0.940 | 0.530 | 0.710 |
| **Remote Sensing Large Vision-Language Models** | | | | | | | |
| GeoChat | 0.255 | 0.470 | 0.105 | 0.455 | 0.660 | 0.810 | 0.647 |
| LHRS-Bot | 0.260 | 0.437 | 0.480 | 0.408 | 0.550 | 0.435 | 0.767 |
| LHRS-Bot-nova | 0.315 | 0.540 | 0.185 | 0.098 | 0.525 | 0.520 | 0.807 |
| VHM | 0.405 | 0.383 | 0.340 | 0.350 | 0.740 | 0.760 | 0.737 |
| RemoteCLIP | 0.335 | 0.210 | 0.035 | 0.245 | 0.820 | 0.650 | 0.740 |
| GeoRSCLIP | 0.400 | 0.310 | 0.310 | 0.260 | 0.945 | 0.935 | 0.880 |

**SII and CID Dimensions.** However, for the perception of finer-grained remote sensing targets, the results in Single-instance Identification and Cross-instance Discernment reveal that the accuracy of most general VLMs falls below 70%, while that of most RSVLMs is under 50%, indicating that current VLMs exhibit limitations in perception capabilities towards more complex remote sensing scenarios and local targets, necessitating further enhancements.

**AttR and AssR Dimensions.** Additionally, capabilities in AttR and AssR, which involve the analysis and reasoning of advanced attributes and features relevant to remote sensing, are largely underdeveloped in most existing VLMs. The evaluated models exhibit an accuracy of nearly 50%, with some RSVLMs even falling below 30%. However, these capabilities are of significant importance and value for the practical applications of large models in authentic remote sensing scenarios, underscoring the need for greater focus in this area.

## 4.3 Fine-grained Analysis

Table 3 and 4 show the experimental results across all L-3 leaf tasks on CHOICE. See more experiments (blind evaluation, Circular evaluation [11], and RES task) in Appendix E.

### 4.3.1 Model Dimensions

**General-domain VLMs match or even outperform RSVLMs in most L-3 tasks.** The superior performance of VHM in SC, OL, and VG demonstrates the efficacy of integrating domain-specific data to bridge the domain gap, but a clear overall disparity remains when compared to general-domain VLMs. Notably, Qwen2-VL-72B achieves the highest accuracy in a majority of perception tasks, exceeding 80% in MR, IC, OL, and CD. InternVL2-40B follows closely, excelling in more complex tasks like AC, TP, and SI, far surpassing RSVLMs. Furthermore, RSVLMs remain confined to the perceptual level, trailing significantly behind general-domain VLMs in reasoning. For instance, while VHM showcases proficiency in Perception tasks, its reasoning capability is notably weaker than that of models with a similar parameter scale, like Ovis1.6-Gemma2-9B, GLM-4V-9B, and Molmo-7B-D. We consistently observe that larger model capacities, exemplified by the Qwen2-VL and InterVL2 series, correlate with enhanced accuracy. These highlight the necessity of remote sensing data and that scaling parameters and incorporating general reasoning capability for RSVLMs can unlock additional performance gains.

Table 4: Fine-grained evaluation results for Perception. Abbreviations adopted: IM for Image Modality; IQ for Image Quality; MR for Map Recognition; SC for Scene Classification; IC for Image Caption; LR for Landmark Recognition; OC for Object Counting; OL for Object Localization; OP for Object Presence; AR for Attribute Recognition; VG for Visual Grounding; HD for Hallucination Detection; AC for Attribute Comparison; SR for Spatial Relationship; CD for Change Detection.

| Model | Image-Level Comprehension | | | | | Single-Instance Identification | | | | | | | Cross-Instance Discernment | | |
|---|---|---|---|---|---|---|---|---|---|---|---|---|---|---|---|
| | IM | IQ | MR | SC | IC | LR | OC | OL | OP | AR | VG | HD | AC | SR | CD |
| **Proprietary Large Vision-Language Models** | | | | | | | | | | | | | | | |
| GPT-4o-mini | 0.781 | 0.500 | 0.490 | 0.937 | 0.940 | 0.940 | 0.542 | 0.612 | 0.914 | 0.604 | 0.000 | 0.906 | 0.426 | 0.340 | 0.650 |
| GPT-4o-2024-11-20 | 0.868 | 0.564 | 0.567 | 0.954 | **1.000** | 0.940 | 0.594 | 0.768 | 0.918 | 0.610 | 0.028 | 0.830 | 0.509 | 0.567 | 0.770 |
| Gemini-1.5-Pro | **0.874** | **0.678** | 0.790 | 0.937 | 0.955 | 0.960 | **0.634** | 0.578 | 0.952 | 0.548 | 0.027 | 0.810 | 0.797 | 0.413 | 0.690 |
| **General-domain Large Vision-Language Models** | | | | | | | | | | | | | | | |
| Qwen2-VL-7B | 0.720 | 0.540 | **0.800** | 0.930 | 0.980 | **0.980** | 0.560 | 0.790 | 0.950 | 0.570 | 0.372 | 0.570 | 0.750 | 0.510 | 0.790 |
| Qwen2-VL-72B | 0.860 | 0.560 | **0.800** | 0.960 | **1.000** | 0.970 | 0.620 | 0.840 | 0.950 | 0.580 | 0.530 | 0.670 | 0.800 | **0.610** | **0.840** |
| Ovis1.6-Gemma2-9B | 0.800 | 0.570 | 0.660 | 0.940 | 0.990 | 0.880 | 0.500 | 0.720 | 0.940 | 0.570 | 0.060 | 0.900 | 0.680 | 0.490 | 0.760 |
| InternVL2-8B | 0.730 | 0.520 | 0.430 | 0.930 | 0.970 | 0.770 | 0.400 | 0.590 | 0.890 | 0.590 | 0.173 | 0.790 | 0.850 | 0.370 | 0.710 |
| InternVL2-26B | 0.780 | 0.540 | 0.610 | 0.950 | 0.960 | 0.980 | 0.460 | 0.680 | 0.960 | 0.620 | 0.052 | 0.730 | 0.660 | 0.410 | 0.760 |
| InternVL2-40B | 0.820 | 0.570 | 0.550 | 0.960 | 0.990 | 0.970 | 0.530 | 0.770 | **0.980** | 0.610 | 0.228 | 0.670 | **0.870** | 0.500 | 0.790 |
| LLaVA-1.6-7B | 0.340 | 0.460 | 0.380 | 0.920 | 0.880 | 0.740 | 0.220 | 0.690 | 0.970 | 0.590 | 0.237 | 0.060 | 0.730 | 0.340 | 0.650 |
| LLaVA-1.6-13B | 0.480 | 0.490 | 0.400 | 0.910 | 0.940 | 0.700 | 0.400 | 0.670 | 0.860 | 0.610 | 0.302 | 0.300 | 0.790 | 0.400 | 0.680 |
| Llama3.2-11B | 0.650 | 0.400 | 0.650 | 0.910 | 0.940 | 0.890 | 0.510 | 0.650 | 0.920 | 0.590 | 0.002 | 0.360 | 0.580 | 0.310 | 0.660 |
| GLM-4V-9B | 0.660 | 0.500 | 0.680 | 0.940 | 0.950 | **0.980** | 0.530 | 0.670 | 0.930 | 0.600 | 0.003 | 0.620 | 0.840 | 0.370 | 0.710 |
| DeepSeek-VL-7B | 0.570 | 0.540 | 0.670 | 0.920 | 0.940 | 0.890 | 0.460 | 0.740 | 0.950 | 0.530 | 0.253 | 0.430 | 0.770 | 0.340 | 0.670 |
| MiniCPM-V-2.5 | 0.610 | 0.430 | 0.690 | 0.930 | 0.980 | 0.900 | 0.460 | 0.600 | 0.940 | 0.610 | 0.055 | 0.640 | 0.790 | 0.400 | 0.700 |
| Phi3-Vision | 0.510 | 0.480 | 0.480 | 0.880 | 0.910 | 0.660 | 0.350 | 0.760 | 0.910 | 0.560 | 0.105 | 0.380 | 0.770 | 0.370 | 0.570 |
| mPLUG-Owl3-7B | 0.770 | 0.420 | 0.670 | 0.890 | 0.960 | 0.920 | 0.500 | 0.660 | 0.950 | 0.590 | 0.073 | 0.380 | 0.520 | 0.300 | 0.710 |
| Molmo-7B-D | 0.560 | 0.530 | 0.350 | 0.870 | 0.920 | 0.740 | 0.390 | 0.630 | 0.920 | **0.670** | 0.015 | 0.760 | 0.800 | 0.490 | 0.620 |
| **Remote Sensing Large Vision-Language Models** | | | | | | | | | | | | | | | |
| GeoChat | 0.313 | 0.299 | 0.433 | 0.922 | 0.710 | 0.790 | 0.218 | 0.762 | 0.934 | 0.510 | 0.297 | 0.064 | 0.671 | 0.300 | 0.415 |
| LHRS-Bot | 0.288 | 0.306 | 0.318 | 0.915 | 0.755 | 0.500 | 0.252 | 0.164 | 0.936 | 0.330 | - | 0.076 | - | 0.267 | 0.325 |
| LHRS-Bot-nova | 0.479 | 0.271 | 0.350 | 0.950 | 0.840 | 0.690 | 0.412 | 0.642 | 0.972 | 0.528 | 0.271 | 0.372 | 0.737 | 0.257 | 0.560 |
| VHM | 0.621 | 0.428 | 0.299 | **0.966** | 0.765 | 0.760 | 0.342 | **0.872** | 0.932 | 0.564 | **0.598** | **0.922** | 0.431 | 0.347 | 0.580 |
| RemoteCLIP | 0.510 | 0.303 | 0.369 | 0.891 | 0.775 | 0.700 | 0.435 | 0.295 | 0.715 | 0.500 | - | 0.050 | 0.688 | 0.248 | 0.545 |
| GeoRSCLIP | 0.705 | 0.303 | 0.471 | 0.944 | 0.860 | **0.980** | 0.230 | 0.360 | 0.750 | 0.250 | - | 0.050 | 0.361 | 0.248 | 0.570 |

**Proprietary VLMs offer no clear advantage in remote sensing tasks.** While three proprietary VLMs demonstrate top-tier capabilities in most L-3 tasks, open-source VLMs like Qwen2-VL-72B and InternVL2-40B show competitive or even superior performance. For example, Qwen2-VL-72B outperforms GPT-4o by 7% in CD and 15.3% in PP, respectively. These open-source VLMs emerge as attractive alternatives, offering a versatile and cost-effective solution for specific tasks.

### 4.3.2 Capability Dimensions

**VLMs struggle with fine-grained perception.** For tasks like OC, AR, and SR, which require local fine-grained information and instance-level interactions, VLMs achieve an accuracy near or below 50%, clearly lower than other perception tasks. Visual grounding, a critical task for target localization in remote sensing, continues to pose significant difficulties, with existing RSVLMs achieving accuracy rates below 60%. General-domain VLMs perform even worse in this task, with some lacking this capability entirely. We attribute these shortcomings to the domain gap between RSIs and natural images, as well as the limitation of VLMs in extracting small-scale objects.

**Reasoning capability is a bottleneck for VLMs.** Critical reasoning tasks for remote sensing applications, such as TP, EA, and RA, remain challenging for most VLMs, with accuracies in these tasks generally falling below 50%, and even below 30% for some RSVLMs. This suggests that current VLMs lack adequate development in reasoning abilities tailored for remote sensing needs.

**General-domain VLMs are more inclined to avoid model hallucination.** In the HD task, we deliberately design problems that prompt VLMs to identify the locations of objects that do not exist in the images [54]. Except for VHM, which has been specifically trained on hallucination data, RSVLMs underperform most general-domain VLMs. However, the refusal capability among general-domain VLMs remains below 70%, which still falls short of the ideal threshold. Moreover, proprietary VLMs tend to favor the "won't wrong" option when uncertain. For instance, GPT-4o selected "can't judge" for 70.2% of the problems in the L-3 RA task. Given the varying resolution and quality of RSIs in practical applications, the refusal for low-quality images and the ability to detect hallucinations deserve greater emphasis from researchers.

## 5 Conclusion and Future Work

In this work, we introduce CHOICE, the first benchmark for systematically and objectively evaluating the remote sensing capabilities of VLMs. Through CHOICE, we assessed 24 mainstream general-

domain VLMs and RSVLMs across hierarchical capability dimensions. Our multi-dimensional and objective analysis reveals the strengths and limitations of these VLMs, providing insights to enhance their effectiveness in remote sensing applications.

Currently, there are also certain limitations of our benchmark. First, in the pixel-level RES task, most VLMs currently lack the ability to handle such problems effectively. As VLMs continue to evolve, we plan to continuously update the evaluation results of the latest models. Second, we tend to incorporate more multi-source data in future work, such as SAR and multi-spectral data. Third, it is evident that the current benchmark cannot fully meet the demand for comprehensiveness. For example, tasks for evaluating urban heat island effect and vegetation health monitoring are absent. Therefore, we will frequently update the benchmark to incorporate a wider variety of related tasks.

## Acknowledgments and Disclosure of Funding

We would like to express our sincere gratitude to all participants in this benchmark for their creativity and dedicated efforts. This work was supported in part by the National Key Research and Development Program of China under Grant 2022YFB3903501 and the National Natural Science Foundation of China under Grant 42271370.

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

# A    Evaluation Dimension

To thoroughly evaluate the perception and reasoning capabilities of VLMs, CHOICE is structured with 6 L-2 dimensions and 23 L-3 leaf tasks. In this section, we provide a detailed definition of each L-3 task, along with a collection of illustrative examples. Following this, we present a detailed statistical summary of CHOICE.

## A.1    Definition of Each L-3 Task

**Image-level Comprehension.** This evaluation dimension contains 5 L-3 tasks, focusing on a broad understanding of image-level content and field-specific relevance. In Figure 4, we present some visualization examples for this dimension. Notably, the titles in **black** indicate that the L-3 tasks are constructed by the label-driven method, titles in **red** signify tasks developed through the foundation model-driven method, and titles in **purple** denote tasks created through human-GPT-4 collaboration.

- Image Modality: Multi-modality is a key feature of remote sensing, with various sensors capturing the distinct types of RSIs, such as optical (RGB), Synthetic Aperture Radar (SAR), false color, and nighttime light. This task aims to assess the VLM's capability to classify the modality of RSIs, which is a fundamental step in selecting the appropriate tools for interpreting RSIs.

- Image Quality: RSIs can be affected by various interference factors during capture, such as cloud cover, different types of noise, and low-light conditions. This task focuses on evaluating the capabilities of identifying the type of interference in RSIs and selecting the RSIs with the highest quality.

- Map Recognition: This task involves assessing the capability of recognizing patterns and shapes across regions of varying scales, as well as evaluating visual world knowledge embedded within VLMs.

- Scene Classification: This task has long been a key downstream application in remote sensing, aiming to classify the category of RSIs using either a closed-set or open-vocabulary approach.

- Image Caption: This task aims to evaluate the VLM's capability to generate coherent and contextually relevant natural language descriptions that accurately capture the overall scene content of RSIs from an aerial perspective.

**Single-instance Identification.** This dimension emphasizes detailed object recognition and localization without reliance on prior knowledge or fixed categories, including 7 L-3 tasks. In Figure 5, we present some visualization examples for this dimension.

- Landmark Recognition: Recognizing the most popular landmarks in the world serves to evaluate the VLM's capability to identify single instances and assess the visual world knowledge embedded within the model.

- Object Counting: This task requires VLMs to accurately identify the objects of interest and then count them within an RSI, which covers a large spatial area.

- Object Localization: Locating the objects of interest is a key challenge in the field of remote sensing. By dividing the RSI into a $3 \times 3$ grid, this task evaluates the VLM's capability for coarse localization of specific instances.

- Object Presence: Determining the presence of a specific instance is a more straightforward task compared to Object Counting and also represents a common practical application in real-world scenarios.

- Attribute Recognition: This task involves identifying an object and recognizing its specific attributes, such as color, making it a highly fine-grained task that assesses the VLM's ability to perceive detailed local information. Due to the need for detailed attribute recognition, the RSI in this task has a finer spatial resolution of 0.1 meter/pixel $\sim$ 0.3 meter/pixel.

- Visual Grounding: As an advanced form of Object Localization, this task focuses on identifying the precise coordinates of the bounding box for a specific instance, similar to traditional object detection tasks. To better evaluate the capability of VLMs, we format

**Image Modality**

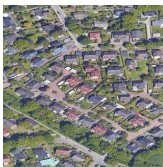

Based on the image provided, which modality does it belong to?
A. false color
B. nighttime light
C. RGB
D. SAR

**Image Quality**

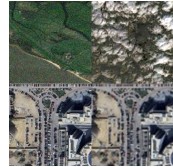

Four images are concatena into a 2 times 2 grid, identify image that has the best quality.
A. bottom left
B. top left
C. bottom right
D. top right

**Scene Classification**

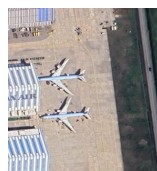

Based on the image provided, which category does it belong to?
A. medium residential
B. dense residential
C. commercial
D. beach

**Map Recognition**

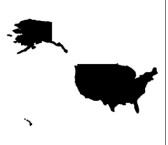

What is the name of the country outlined in this map?
A. Mauritania
B. Congo
C. United Kingdom
D. United States of America

**Image Caption**

This is a remote sensing image of a airport scene. Please describe this remote sensing image in one shot sentence?
A. A high-altitude image of a stadium with four event tents set up in the field, bordered by various support vehicles and structures.
B. A bird's-eye view of a railway station featuring three trains stationed on the tracks, encircled by numerous service vehicles and facilities.
C. An aerial view of an airport terminal with two airplanes parked at the gates, surrounded by various ground vehicles and infrastructure.
D. An overhead perspective of a seaport with several ships docked at the piers, surrounded by multiple loading trucks and equipment.

Figure 4: Examples of L-3 tasks within the L-2 dimension of image-level comprehension.

this task as free-form questions, prompting VLMs to output the bounding box coordinates directly in the format of $[x_1, y_1, x_2, y_2]$. The mIoU metric is computed with the correct label, with a score above 0.5 considered a correct detection.

- Hallucination Detection: Model hallucination poses a significant challenge for VLMs. In this task, we utilize the attributes of specific instances to assess the capability of VLMs in detecting and mitigating hallucinations.

**Cross-instance Discernment.** This dimension emphasizes the comparisons or changes among multiple objects of interest, which provides insights into contextual dependencies, spatial dynamics, and temporal changes within an image. Three L-3 tasks are included, and Figure 6 presents some visualization examples for each task.

- Attribute Comparison: This task is aimed at comparing the attributes, such as color, between instances to identify similarities and differences in their visual characteristics.
- Spatial Relationship: Building on the locations of individual instances within an RSI, this task focuses on analyzing the spatial relationships between these instances to understand patterns and interactions among them.
- Change Detection: Analyzing temporal changes between two paired RSIs is an essential capability necessary for VLMs. This task requires the VLMs to describe the changed areas in natural language.
- Referring Expression Segmentation: This task involves accurately segmenting and identifying specific regions within an RSI based on a natural language description. It challenges VLMs to translate textual descriptions into pixel-wise segmentation masks.

**Attribute Reasoning.** In this dimension, VLMs are required to reason about the overall characteristics of RSIs or the specific attributes of the instances. This dimension includes 2 Level-3 tasks, with visualization examples for each task shown in Figure 7.

**Landmark Recognition**

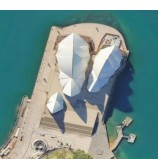

Based on the image provided, which landmark is depicted?
A. Kremlin
B. Machu Picchu
C. Tsarskoye Selo
D. Sydney Opera House

**Object Counting**

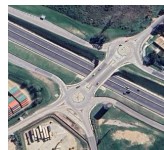

How many roundabout are there in this image?
A. 1
B. 2
C. 4
D. 3

**Object Localization**

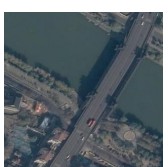

Where is the bridge located in this image??
A. Right
B. Center
C. Bottom
D. Bottom Left

**Object Presence**

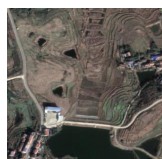

Is there a/an dam in this image?
A. Yes
B. No
C. Can't Judge

**Attribute Recognition**

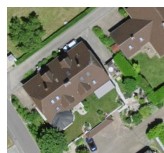

What color is the vehicle in the Bottom of this image?
A. blue
B. red
C. yellow
D. brown

**Visual Grounding**

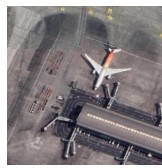

Please output the coordinates of the airplane in the image in the format (x1, y1, x2, y2).
Label: [0.52, 0.18, 0.71, 0.59]

**Hallucination Detection**

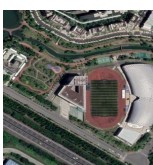

Where is the swimming-pool located in this image.
A. Top Left
B. Top
C. Bottom Left
D. The object is not in the picture

Figure 5: Examples of L-3 tasks within the L-2 dimension of single-instance identification.

- Time Property. This task highlights the VLM's capability to deduce the season in which the RSI was captured based on the overall visual information depicted in the image.

- Physical Property: This task focuses on attributes of instances that require real-world measurements, such as estimating the height of a building depicted in the RSI.

**Assessment Reasoning.** In this dimension, VLMs are required to reason about estimated indicators of societal development and environmental conditions depicted in the RSI. This dimension includes 2 Level-3 tasks, with visualization examples for each task shown in Figure 8.

- Environmental Assessment: This task evaluates a VLM's capability to derive meaningful and high-level insights into environmental information, such as the $CO_2$ emissions within an area depicted in the RSI.

- Resource Assessment: This task evaluates a VLM's capability to deduce the estimated resource quantity present in a given area, such as the population sizes and economic situations.

**Common Sense Reasoning.** In this dimension, VLMs are required to reason about real-world scenarios based on the extensive common-sense knowledge embedded in LLM. This dimension includes 3 Level-3 tasks, with visualization examples for each task shown in Figure 9.

- Disaster Discrimination: Recognizing types of natural disasters is a fundamental aspect of common sense knowledge. This task evaluates the VLM's capability to identify and differentiate among various natural disasters based on visual cues within the RSIs.

- Geospatial Determination: Identifying landmarks and reasoning about their geographic locations, such as determining the city or country, is another component of common sense

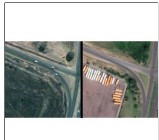

Are the two vehicles in this image the same color?
A. Yes
B. No
C. Can't Judge

**Spatial Relationship**

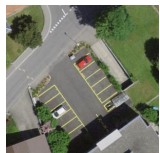

In this picture, what is the position of the White vehicle in relation to the Red vehicle?
A. Top
B. Top left
C. Bottom Left
D. Top Right

**Change Detection**

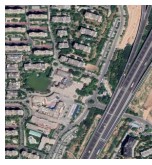

What changes have occurred from the left scene to the right scene?
A. A parking lot was built on the wasteland, with several large vehicles neatly parked inside.
B. An office complex was constructed on the empty terrain, with numerous compact trucks arranged in an orderly fashion.
C. A playground was established on the deserted ground, filled with vibrant bicycles lined up systematically.
D. A storage facility was erected on the barren land, featuring a variety of small cars organized in rows.

**Referring Expression Segmentation**

Original Image    Give me the mask of the bigger roundabout in this image.    Segment the top roundabout shown in this image.

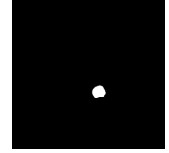

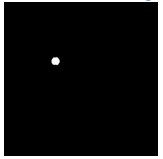

Figure 6: Examples of L-3 tasks within the L-2 dimension of cross-instance discernment.

**Time Property**

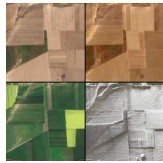

In the 2×2 grid of seasonal images of the same area, identify the one that was captured in winter.
A. Bottom Left
B. Top right
C. Bottom right
D. Top left

**Physical Property**

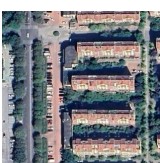

Which of the following options is closest to the height in meters of the building depicted in the image?
A. 17
B. 105
C. 87
D. 95

Figure 7: Examples of L-3 tasks within the L-2 dimension of attribute reasoning.

**Environmental Assessment**

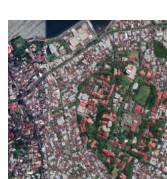

From this image, which carbon dioxide emission level is most representative of this area?
A. High
B. Very Low
C. Medium
D. Low

**Resource Assessment**

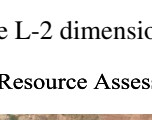

What is the estimated population living in this area shown in the image?
A. 146944
B. 163315
C. 114165
D. 2616

Figure 8: Examples of L-3 tasks within the L-2 dimension of assessment reasoning.

knowledge. This task assesses the VLM's capability to recognize landmarks and associate them with their geographic contexts.

- Situation Inference: In the real world, knowing what items to carry when visiting different locations or understanding the functions of specific buildings is common-sense knowledge that humans naturally possess. This task assesses the VLM's capability to infer situational context based on scene understanding within the RSIs.

**Disaster Discrimination**

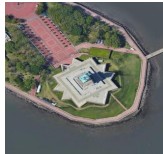

What disaster led to the situation depicted in this image?
A. Flood
B. Earthquake
C. Fire
D. Landslide

**Geospatial Determination**

Based on the image provided, which country is this landmark located in?
A. France
B. Egypt
C. USA
D. UK

**Situation Inference**

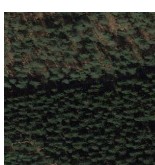

Which animal is most likely to appear in the scene in the picture?
A. seahorse
B. Bear
C. Whale
D. dolphin

Figure 9: Examples of L-3 tasks within the L-2 dimension of common sense reasoning.

Table 5: The detailed statistics for CHOICE. Abbreviations adopted: MCQ for Multi-Choice Question; FC for False Color; NL for Nighttime Light; MI for Multi-Image; BT for Bi-Temporal; MT for Multi-Temporal.

| Level-1 | Level-2 | Level-3 | Data Source | Answer Type | Modality | Resolution | Image Size | # Sample |
|---|---|---|---|---|---|---|---|---|
| Perception | ILC | IM | Google Earth Engine Landsat-8 SDGSAT-1 Sentinel-1 | 4-option MCQ | RGB SAR FC NL | 0.3m~30m | 512×512 | 800 |
| | | IQ | Google Earth Engine Landsat-8 | 3/4-option MCQ | RGB MI | 0.3m~30m | 512×512 | 800 |
| | | MR | Natural Earth | 4-option MCQ | RGB | N/A | 512×512 | 157 |
| | | SC | Google Earth Engine | 4-option MCQ | RGB | 0.3m~10m | 512×512 | 2000 |
| | | IC | Google Earth Engine | 4-option MCQ | RGB | 0.3m | 512×512 | 200 |
| | SII | LR | Google Earth Engine | 4-option MCQ | RGB | 0.3m | 512×512 | 100 |
| | | OC | Google Earth Engine | 4-option MCQ | RGB | 0.3m | 800×800 | 500 |
| | | OL | Google Earth Engine | 4-option MCQ | RGB | 0.3m | 800×800 | 500 |
| | | OP | Google Earth Engine | 3-option MCQ | RGB | 0.3m | 800×800 | 500 |
| | | AR | SWISSIMAGE[1] | 4-option MCQ | RGB | 0.1~0.3m | 800×800 | 500 |
| | | VG | Google Earth Engine | Bounding Box | RGB | 0.3m | 512×512 | 600 |
| | | HD | Google Earth Engine | 4-option MCQ | RGB | 0.3m | 800×800 | 500 |
| | CID | AC | SWISSIMAGE | 3-option MCQ | RGB | 0.1~0.3m | 800×800 | 350 |
| | | SR | SWISSIMAGE | 4-option MCQ | RGB | 0.1~0.3m | 800×800 | 300 |
| | | CD | Google Earth Engine | 4-option MCQ | RGB, BT | 0.3m | 512×512 | 200 |
| | | RES | Google Earth Engine | Seg. Mask | RGB | 0.3m | 800×800 | 500 |
| Reasoning | AttR | TP | Sentinel-2 | 4-option MCQ | RGB, MT | 10m | 512×512 | 200 |
| | | PP | Google Earth Engine CNBH | 4-option MCQ | RGB | 0.1m~0.3m | 512×512 | 300 |
| | AssR | EA | Google Earth Engine ODIAC | 4-option MCQ | RGB | 2.4m | 512×512 | 200 |
| | | RA | Google Earth Engine WorldPop[2] | 3/4-option MCQ | RGB MI | 10m | 512×512 | 600 |
| | CSR | DD | Diffusion Model | 4-option MCQ | RGB | N/A | 512×512 | 200 |
| | | GD | Google Earth Engine | 4-option MCQ | RGB | 0.3m | 512×512 | 200 |
| | | SI | Google Earth Engine | 4-option MCQ | RGB | 0.3m~10m | 512×512 | 300 |
| **Total** | | | | | | | | **10507** |

## A.2 Statistics of CHOICE

All RSIs in CHOICE are sourced from a variety of satellites, products, and platforms. To ensure a balanced evaluation, we targeted an even distribution among problems related to different capabilities during data collection, with a minimum of 150 samples per task. The data sources and the number of problems are listed in Table 5.

## B    Data Leakage in Current Benchmarks

As shown in Table 1, the majority of mainstream multi-modal datasets and benchmarks in the field of remote sensing are repurposed from commonly used datasets [18, 19], some of which, such as

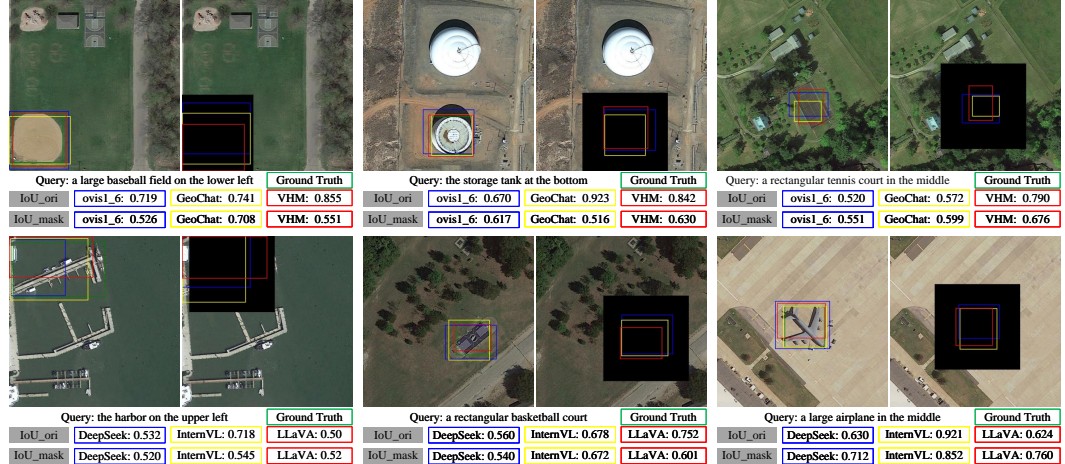

Figure 10: Examples of data leakage in DIOR-RSVG dataset. Model abbreviations adopted: Ovis1_6 for Ovis1.6-Gemma2-9B; DeepSeek for DeepSeek-VL-7B; InternVL for InternVL2-8B; LLaVA for LLaVA-1.6-7B.

Table 6: The number of cities per continent in the Oxford Economics' Global Cities Index and the number of cities randomly selected for data collection.

| Continent | Asia | Europe | North America | Africa | South America | Oceania | **Total** |
|---|---|---|---|---|---|---|---|
| # Cities | 441 | 218 | 149 | 113 | 67 | 12 | **1000** |
| # Selected Cities | 22 | 11 | 7 | 6 | 3 | 1 | **50** |

DOTA [20] and DIOR [21], have been utilized during the training stages of VLMs. This practice of data repurposing inevitably leads to data leakage, which compromises the objectivity of the evaluation process. We use DIOR-RSVG [17], which is further processed from the object detection dataset DIOR [21], as an example to demonstrate such data leakage in the context of the visual grounding task. Specifically, one-quarter of the image that contains the object of interest is masked while the query remains unchanged. We then compare the outputs of the VLMs with those obtained from the original images. As illustrated in Figure 10, the results inferred from the masked image still yield an IoU similar to that of the intact images. This reflects the prior knowledge of VLMs on responding even without visual evidence and highlights the inherent data leakage. Therefore, most existing benchmarks derived from such common datasets fail to accurately and objectively reflect the true proficiency of VLMs in remote sensing tasks.

## C Details on CHOICE Construction

To increase data diversity, most RSIs in CHOICE are sourced from 50 randomly selected cities across six continents (excluding Antarctica), based on the top 1,000 global cities identified by Oxford Economics' Global Cities Index, minimizing potential biases in the data. The details of data coverage are listed in Table 6.

We employ three approaches to collect RSIs and create associated questions: (1) label-driven construction, (2) foundation model-driven construction, and (3) human-GPT4 collaboration. In this section, we provide a more detailed construction process for each L-3 leaf task.

For Scene Classification, we begin by defining a set of categories for RSIs to guide data collection. OpenStreetMap (OSM) represents physical features on the ground (e.g., roads or buildings) using tags (i.e., key-value pairs), with each tag describing a geographic attribute of the feature. For example,

---

[1] https://www.swisstopo.admin.ch/de/en/orthoimage-swissimage-10
[2] https://www.worldpop.org/

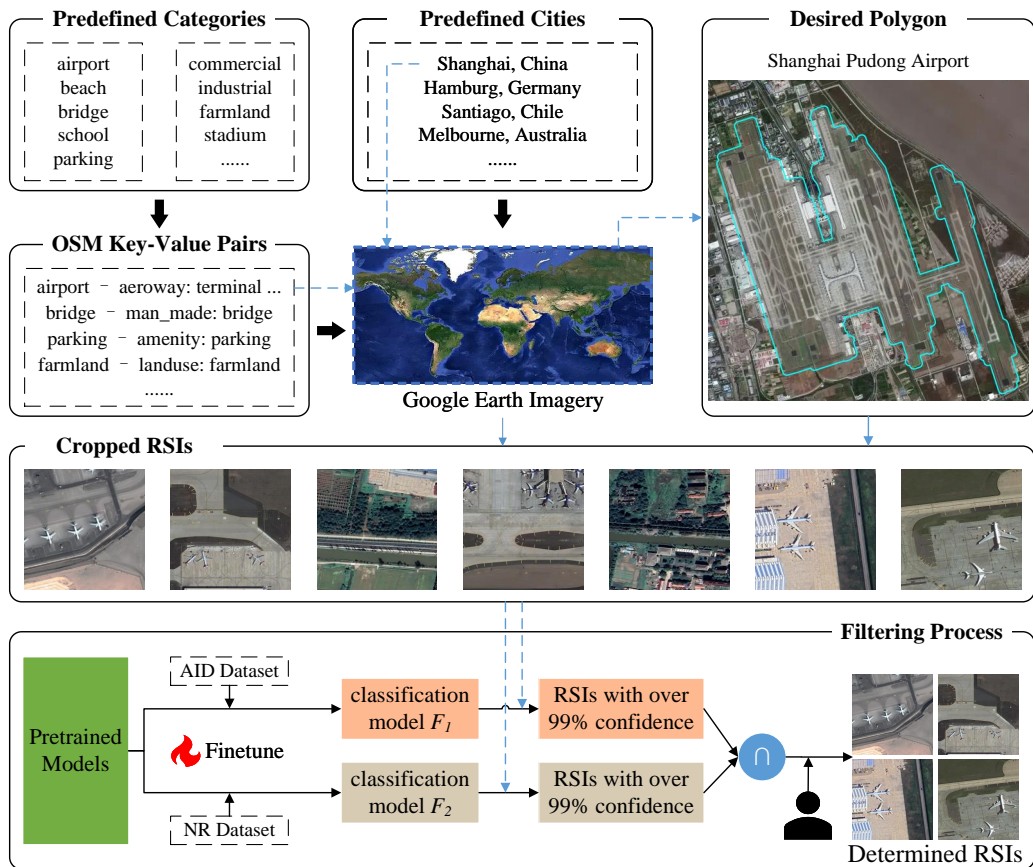

Figure 11: Overview of the construction procedure for the Scene Classification task.

an aerodrome, airport or airfield can be tagged as $aeroway = aerodrome$. Taking the common categories in existing relevant datasets [55, 56] as a reference, we follow the OSM's tagging system to establish a category set, as shown in Table 8, with each predefined category can be described by one or more key-value pairs. We further select 10 of the 50 chosen cities as the primary data collection areas to capture more intra-class variations, including Sapporo, Mecca, Busan, Shanghai, Hamburg, Amsterdam, Phoenix, Cairo, Santiago, and Melbourne. As shown in Figure 11, we obtain polygons representing the locations of interest based on the predefined key-value pairs for each category. Using these polygons, we then download the corresponding RSIs from Google Earth Engine. To ensure high-quality data, we design an automatic filtering pipeline to exclude the low-confidence RSIs resulting from the inaccuracy in OSM. Following [14], we finetune two versions of classification models based on the 30-category AID dataset and 45-category NR dataset, both of which include the 20 categories we defined. After cropping the RSIs we collect into $512 \times 512$ tiles, we input them into the two models and take the intersection of the outputs with over 99% confidence from both models as the desired RSIs. Finally, human annotators review the results to further ensure the accuracy of all categories. We ultimately collect 2,000 high-confidence RSIs across 20 categories for this task.

For Image Quality, we first define 4 common types of noise that typically interfere with RSIs: Gaussian noise, impulse noise, stripe noise, and deadline noise. We randomly select 200 samples in Scene Classification evenly from the original high-quality RSIs, and manually add each type of noise to these samples, creating 50 samples for each noise type, resulting in 200 multiple-choice questions (MCQs) for identifying the type of noise present in an RSI. Moreover, we concatenate each noisy RSI with its corresponding original RSI side by side, forming 200 two-option MCQs to determine which version has better quality. We further collect cloudy RSIs from the Landsat-8 satellite, which has a spatial resolution of 30 meters/pixel, across the 10 cities. By selecting tiles with 60% cloud coverage and cropping the corresponding areas into $512 \times 512$ patches, we combine Bands 4, 3, and 2 to obtain the natural color images, which results in 200 cloudy RSIs. We then concatenate three RSIs

> **Prompts for GPT4 to Generate Distracting Choices**
>
> **Image Caption:**
> "Using the provided sentence as a reference description of an aerial image, generate three unique sentences with distinct meanings. Each sentence should reinterpret the scene by altering key aspects such as:
> - objects: replace primary objects (e.g., buildings, trees, water bodies) with other objects that might be found in an aerial image.
> - spatial relationships: modify directional or spatial terms (e.g., replace "top of" with "bottom of" or change proximity descriptions).
> - visual attributes: change details related to color, texture, or shape (e.g., "green trees" to "golden fields").
> - scene context: adjust the overall scene by subtly shifting the setting or focus, such as moving from a rural to an urban scene or vice versa.
>
> Ensure that each sentence describes a plausible yet different interpretation of the aerial image, so they each stand on their own with varied imagery and meaning."
>
> **Change Detection:**
> "Based on the given sentence that describes changes observed between two remote sensing images captured at different timestamps, create three new sentences with varied meanings. Each sentence should reimagine the scene by:
> - substituting main features: change key objects or elements, such as replacing "buildings" with "forest areas" or "open fields".
> - altering time-based descriptions: describe the changes differently over time, focusing on aspects like expansion, reduction, or relocation of elements.
> - modifying spatial relationships: adjust terms that indicate direction or proximity, such as using "near" instead of "at the top of" or emphasizing different spatial arrangements.
> - varying color and texture descriptions: change color references or descriptive qualities, such as "lush green" to "arid yellow" or "dense" to "sparse".
>
> Each sentence should uniquely reflect the scene with these modifications, ensuring all descriptions provide distinct interpretations of the changes observed."

Figure 12: The prompts used for generating distracting descriptions in Image Caption and Change Detection tasks.

from Time Property and one cloudy RSI into a $2 \times 2$ grid, and generate 200 MCQs for identifying the cloudy RSI. To comprehensively assess the VLM's capability to identify and compare the quality of RSIs, we concatenate the original RSI, its corresponding noisy version, the cloudy RSI and the low-resolution RSI (obtained by randomly cropping the original RSI and resizing it to the original size) into a $2 \times 2$ grid, forming 200 MCQs to identify the best quality among the four options.

For Image Modality, we include four modalities of RSIs: RGB, SAR, false color, and nighttime light. We still select the RSIs from Scene classification as the RGB group, and combine Bands 5, 4, and 3 of the cloudy RSIs from Landsat-8 as the false color modality. With the assistance of the Sentinel-1 satellite with a spatial resolution of 10 meters/pixel, we download and crop SAR RSIs (both VV and VH polarizations) across the 10 cities via AI Earth[3]. As for nighttime light images, we manually choose RSIs from the SDGSAT-1 satellite. We then randomly select 200 RSIs per modality to form 800 MCQs for identifying the modality of each RSI.

For Map Recognition, we first obtain the maps of the 7 continents. From Oxford Economics' Global Cities Index, which includes 1,000 cities across 162 countries, we randomly select 100 countries and 50 cities. We then source maps for these selected locations from Natural Earth.

For Image Caption, we randomly select 200 RSIs from Scene Classification, ensuring an even distribution, and employ human annotators to generate a descriptive caption for each RSI. Then we prompt GPT-4 to generate 3 distracting descriptions, with these descriptions subsequently verified by human reviewers to ensure accuracy and relevance. The prompt we use is shown in Figure 12.

For Landmark Recognition, we choose 100 most famous landmarks in the world from List Challenges[4] and download the RSIs with 0.3 meter/pixel spatial resolution from Google Earth Engine according to their latitude and longitude coordinates.

---

[3] https://engine-aiearth.aliyun.com
[4] https://www.listchallenges.com/

For the other 6 leaf tasks in the Single-Instance Identification dimension, objection detection for identifying instances and obtaining their coordinates of bounding boxes is the first step to generate associated problems. As mentioned in Scene Classification, we also predefine a set of categories for instances based on the existing object detection datasets [20, 21] and OSM key-value pairs, as shown in Table 9. We select 30 of the 50 chosen cities as the primary data collection areas to capture more intra-class variations, including Riyadh, Gwangju, Tokyo, Daejeon, Chengdu, Istanbul, Nagoya, Wuhan, Hsinchu, Nanjing, Kuala Lumpur, Jeddah, Lyon, Paris, Rotterdam, Brussels, Berlin, Munich, Vienna, Zurich, Atlanta, Boston, Washington D.C., Montreal, Calgary, Windhoek, Gaborone, Rabat, Montevideo and Lima. Using the same procedure as for Scene Classification, we obtain the original RSIs from Google Earth Engine and crop them into $800 \times 800$ patches. We apply two object detection models, MTP [40] pretrained on DIOR-R [57] and YOLOv8 [41] pretrained on DOTA [20], to automatically generate bounding boxes for the predefined categories. Human annotators are then employed to verify the completeness and accuracy of these labels, retaining only those with high confidence from both models. Once getting the labels and coordinates of these instances, we count the number of specific objects for Object Presence and Object Counting. We compute the centroids of these objects and assign their coarse locations in a $3 \times 3$ grid for Object Localization, and use the normalized coordinates of bounding boxes directly as the label for Visual Grounding. For Hallucination Detection, we aim to assess the VLM's capability to realize the non-existent instances within RSIs and correctly refuse to respond to such queries.

For Attribute Recognition, RSIs collected from Google Earth Engine with a resolution of 0.3 meter/pixel are not sufficient for fine-grained attributes such as vehicle color. Therefore, we collect additional RSIs with a higher resolution of 0.1 meter/pixel. We then utilize K-means clustering to classify the color of the vehicles and obtain the direction by the angle of bounding box, generating MCQs related to these attributes.

For Attribute Comparison, building upon Attribute Recognition, we generate MCQs that compare the similarity or difference of specific attributes between instances within an RSI.

For Spatial Relationship, we select RSIs that contain two instances with different attributes and generate MCQs to identify the spatial relationship between them, such as bottom left and top right.

For Change Detection, we collect registered RSIs captured at two different timestamps from 6 of the 50 cities, including Nicosia, Haifa, Dammam, Malmo, Seattle, and Pretoria. Human annotators are enlisted to generate a descriptive caption for each pair of RSIs that highlights the temporal changes between them. GPT-4 is then prompted to generate 3 distracting options, with these descriptions subsequently verified by human reviewers to ensure accuracy and relevance. The prompt we use is shown in Figure 12.

For Referring Expression Segmentation, we follow the method mentioned in Single-instance Identification. Instead of applying object detection foundation models, we employ a segmentation model, RingMo-SAM [42], to generate precise segmentation masks for the objects of interest, with the bounding boxes serving as the visual prompts to guide the segmentation process, ensuring more accurate delineation of the objects. By computing and comparing the locations or areas of these object masks, we are able to automatically construct RES problems, such as "Give me the segmentation mask of the larger roundabout."

For Time Property, we follow the method in [58] to collect four seasonal RSIs from 4 of the 50 cities: Kanazawa, Hong Kong, Bangkok, and Qingdao. Volunteers are employed to select timestamps that clearly represent each season. 200 MCQs are generated to assess the ability to identify the season in which each RSI was captured.

For Physical Property, we utilize the CNBH-10m [59] product, which provides the estimated building heights in China, to guide the collection of RSIs. We randomly select 100 points from this product and download the corresponding RSIs from Google Earth Engine. To ensure accuracy, volunteers compare the estimated building heights provided by CNBH-10m with human-assessed heights from the RSIs. Any discrepancies are corrected by human annotators to maintain accuracy in building height measurements. Moreover, we incorporate an additional 200 MCQs to determine whether vehicles are stationary or in motion, based on the images collected in Attribute Recognition.

For Environmental Assessment and Resource Assessment, we use the ODIAC Fossil Fuel Emission Dataset and WorldPop Estimated Residential Population, respectively, and then follow the same procedure as Physical Property to generate associated MCQs.

Figure 13: System prompts used for different RSVLMs or tasks, and an example for evaluating CLIP-based VLMs on MCQs.

For Disaster Discrimination, due to the challenges in collecting disaster-related RSIs, we prompt a diffusion model[5] to generate synthetic RSIs depicting four types of disasters: earthquake, fire, flood, and landslide. For each disaster type, we generate 50 samples, resulting in a total of 200 disaster-related RSIs. These generated RSIs are then used to assess the VLM's capability to identify and differentiate between these types of natural disasters.

For Geospatial Determination, we utilize the RSIs from Landmark Recognition and obtain the corresponding city and country for each landmark. 200 MCQs are generated to identify the cities and countries to which these landmarks belong, relying on common sense and geographic knowledge.

For Situation Inference, we use a subset of RSIs from Scene Classification to generate MCQs that assess the capability to reason about behaviors or features within specific daily situational contexts.

# D   Details on Evaluation

Table 7 provides details of all open-source VLMs evaluated in CHOICE. By leveraging the instruction-following ability of VLMs, we restrict the model response to either the A/B/C/D option or the corresponding content of the choice. This constraint simplifies subsequent choice extraction and enables straightforward accuracy computation. All open-source general-domain VLMs evaluated in CHOICE are reproduced within the ms-swift framework [60], employing their default generation configurations. All inference experiments are performed on up to 8 NVIDIA 3090 GPUs, depending on the parameter size of the VLM. As shown in Figure 13, we design a system prompt tailored for both general-domain VLMs and RSVLMs when handling tasks with MCQs. For Visual Grounding, where the response requires free-form coordinates, we retain the original system prompts specific to RSVLMs, as these are better suited for generating precise coordinate outputs. This dual approach

---

[5]https://civitai.com/models/6424/chilloutmix

Table 7: Details of the evaluated open-source VLMs.

| Model | Vision Backbone | Language Backbone | Overall Parameters |
|---|---|---|---|
| **General-domain Large Vision-Language Models** | | | |
| Qwen2-VL-7B | ViT-L | Qwen2-7B | 7B |
| Qwen2-VL-72B | ViT-L | Qwen2-72B | 72B |
| Ovis1.6-Gemma2-9B | Siglip-400M | Gemma2-9B-It | 9B |
| InternVL2-8B | InternViT-300M-448px | internlm2_5-7b-chat | 8.1B |
| InternVL2-26B | InternViT-6B-448px-V1-5 | internlm2-chat-20b | 25.5B |
| InternVL2-40B | InternViT-6B-448px-V1-5 | Nous-Hermes-2-Yi-34B | 40.1B |
| LLaVA-1.6-7B | CLIP ViT-L/14 | Vicuna-v1.5-7B | 7.1B |
| LLaVA-1.6-13B | CLIP ViT-L/14 | Vicuna-v1.5-13B | 13.4B |
| Llama3.2-11B | - | Llama 3.1 | 10.6B |
| GLM-4V-9B | - | - | 9B |
| DeepSeek-VL-7B | SAM-B&SigLIP-L | DeekSeek-7B | 7.3B |
| MiniCPM-V-2.5 | Siglip-400M | Llama3-8B-Instruct | 8.5B |
| Phi3-Vision | CLIP ViT-L/14 | Phi-3-mini | 4.2B |
| mPLUG-Owl3-7B | Siglip-400M | Qwen2-7B | 8B |
| Molmo-7B-D | CLIP ViT-L/14-336px | Qwen2-7B | 7B |
| **Remote Sensing Large Vision-Language Models** | | | |
| GeoChat | CLIP ViT-L/14-336px | Vicuna-v1.5-7B | 7B |
| LHRS-Bot | CLIP ViT-L/14 | LLaMA2-7B | 7B |
| LHRS-Bot-nova | SigLIP-L/14-336px | LLaMA3-8B | 8B |
| VHM | CLIP ViT-L/14-336px | Vicuna-v1.5-13B | 7B |

ensures optimal performance across varied task requirements while maintaining consistency and accuracy in evaluation.

We also present an example of the evaluation strategy for CLIP-based VLMs in Figure 13. The MCQ is transformed into $n$ declarative sentences by adding or replacing keywords in the question and further rearranging the word order. During the evaluation stage, the similarity score between the RSI and each sentence is computed. The choice corresponding to the sentence that achieves the highest similarity score is considered the model's response. Notably, CLIP-based VLMs lack the capability for visual grounding, and directly computing similarity scores between RSIs and pure coordinates is not feasible. Consequently, this leaf task is left unaddressed for CLIP-based VLMs.

**Failure Cases.** For Visual Grounding, the responses from LHRS-Bot consistently fall within a limited set of repeated outputs across all problems, such as $(0.44, 0.44, 0.52, 0.52)$, $(0.34, 0.44, 0.66, 0.66)$, and $(0.38, 0.38, 0.59, 0.59)$. This pattern results in extremely low accuracy and highlights a significant lack of diversity and adaptability in the model's predictions for this task. For Attribute Comparison, LHRS-Bot consistently outputs the choice "No", regardless of how the choice order is altered. This behavior indicates a fundamental lack of capability in the model to perform attribute comparison tasks effectively.

Table 8: The predefined categories of scene classification and associated OSM key-value pairs

| Category | OSM-Keys | OSM-Values |
|---|---|---|
| airport | aeroway | apron; terminal; aerodrome |
| baseball_field | leisure sports | pitch baseball |
| beach | nature building leisure | beach beach_nut beach_resort |
| bridge | man_made | bridge |
| church | building landuse | church; synagogue; cathedral religious |
| commercial | building landuse | commercial commercial |
| dense_residential medium_residential sparse_residential | building landuse place | residential; cabin; bungalow; detached; farm; house residential farm; isolated_dwelling |
| desert | nature | dune; sand |
| farmland | landuse | farmland |
| forest | landuse | forest |
| industrial | building landuse | industrial industrial |
| parking | amenity | parking |
| pind | water | pond |
| river | waterway | river |
| roundabout | junction | roundabout |
| school | amenity building | school; college; university education |
| stadium | building leisure | stadium stadium |
| storage_tanks | building man_made | storage_tank storage_tank |

Table 9: The predefined categories of instance-wise tasks and associated OSM key-value pairs

| Category | OSM-Keys | OSM-Values |
|---|---|---|
| airplane | aeroway | apron; terminal; aerodrome |
| baseball_field | leisure sports | pitch baseball |
| basketbal_field | leisure sports | pitch basketball |
| soccer_ball_field | leisure sports | pitch soccer |
| tennis_court | leisure sports | pitch tennis |
| bridge | man_made | bridge |
| dam | waterway | dam |
| ground_track_field | leisure | playground |
| harbor | industrial | port |
| ship | industrial | port |
| storage_tank | building man_made | storage_tank storage_tank |
| roundabout | junction | roundabout |

# E  More Experiment

## E.1  Blind Evaluation

We claim to ensure the objectivity of our evaluation based on CHOICE by, on one hand, preventing data leakage, which is accomplished by collecting all data on our own from diverse platforms, and on the other hand, strictly ensuring that the correct answer can only be derived based on a thorough understanding of visual content. As pointed out in MMStar [13], visual content may be unnecessary for many evaluation samples, in which case the assessment of VLMs' multi-modal capabilities degrades to merely evaluating the uni-modal performance of their LLM backbones. Therefore, we conducted a blind evaluation, where VLMs are required to answer the question without visual RSI input (considering the costs, we only conducted experiments on open-source models). We expect that the results of VLMs across all tasks are similar to random selection (i.e., randomly choosing an answer as a response).

Table 10, 11, and 12 present the outcomes of blind evaluation. To our expectation, the results for both L-2 and fine-grained L-3 tasks are close to the random selection, demonstrating the dependence on RSIs to answer the questions and confirming the reliability of our multi-modal benchmark. Notably, in the absence of visual input, some VLMs tend to respond with the refusal choice, such as "can't judge" in Attribute Comparison (AC) and "The object is not in the picture" in Hallucination Detection (HD). Therefore, the high performances of the Qwen2-VL series, InternVL2 series, and VHM inthe HD task of blind evaluation further demonstrate their strong HD capability, which aligns with the vanilla results in Table 4.

Table 10: Random and blind evaluation results for L-2 dimensions. Abbreviations adopted: ILC for Image-level Comprehension; SII for Single-instance Identification; CID for Cross-instance Discernment; AttR for Attribute Reasoning; AssR for Assessment Reasoning; CSR for common sense Reasoning.

| Model | ILC | SII | CID | AttR | AssR | CSR |
|---|---|---|---|---|---|---|
| Random | 0.267 | 0.209 | 0.261 | 0.284 | 0.298 | 0.273 |
| **General-domain Large Vision-Language Models** | | | | | | |
| Qwen2-VL-7B | 0.253 | 0.299 | 0.292 | 0.238 | 0.153 | 0.246 |
| Qwen2-VL-72B | 0.280 | 0.304 | 0.178 | 0.142 | 0.215 | 0.290 |
| Ovis1.6-Gemma2-9B | 0.278 | 0.321 | 0.309 | 0.324 | 0.300 | 0.266 |
| InternVL2-8B | 0.262 | 0.355 | 0.286 | 0.280 | 0.290 | 0.320 |
| InternVL2-26B | 0.244 | 0.343 | 0.261 | 0.156 | 0.163 | 0.263 |
| InternVL2-40B | 0.270 | 0.283 | 0.318 | 0.244 | 0.075 | 0.296 |
| LLaVA-1.6-7B | 0.273 | 0.220 | 0.304 | 0.298 | 0.305 | 0.261 |
| LLaVA-1.6-13B | 0.268 | 0.258 | 0.341 | 0.332 | 0.268 | 0.277 |
| Llama3.2-11B | 0.219 | 0.234 | 0.157 | 0.214 | 0.173 | 0.293 |
| GLM-4V-9B | 0.275 | 0.257 | 0.328 | 0.312 | 0.345 | 0.223 |
| DeepSeek-VL-7B | 0.262 | 0.209 | 0.311 | 0.310 | 0.288 | 0.307 |
| MiniCPM-V-2.5 | 0.284 | 0.293 | 0.279 | 0.322 | 0.338 | 0.223 |
| Phi3-Vision | 0.289 | 0.310 | 0.306 | 0.188 | 0.175 | 0.209 |
| mPLUG-Owl3-7B | 0.233 | 0.262 | 0.231 | 0.194 | 0.165 | 0.304 |
| Molmo-7B-D | 0.247 | 0.267 | 0.319 | 0.310 | 0.378 | 0.286 |
| **Remote Sensing Large Vision-Language Models** | | | | | | |
| GeoChat | 0.251 | 0.242 | 0.275 | 0.332 | 0.353 | 0.241 |
| LHRS-Bot | 0.276 | 0.172 | 0.165 | 0.320 | 0.375 | 0.243 |
| LHRS-Bot-nova | 0.258 | 0.270 | 0.287 | 0.198 | 0.133 | 0.291 |
| VHM | 0.255 | 0.390 | 0.289 | 0.248 | 0.100 | 0.240 |

Table 11: Fine-grained random and blind evaluation results for Perception. Abbreviations adopted: IM for Image Modality; IQ for Image Quality; MR for Map Recognition; SC for Scene Classification; IC for Image Caption; LR for Landmark Recognitionl; OC for Object Counting; OL for Object Localization; OP for Object Presence; AR for Attribute Recognition; HD for Hallucination Detection; AC for Attribute Comparison; SR for Spatial Relationship; CD for Change Detection.

| Model | Image-Level Comprehension | | | | | Single-Instance Identification | | | | | | | Cross-Instance Relationship | | |
|---|---|---|---|---|---|---|---|---|---|---|---|---|---|---|---|
| | IM | IQ | MR | SC | IC | LR | OC | OL | OP | AR | VG | HD | AC | SR | CD |
| Random | 0.260 | 0.290 | 0.290 | 0.260 | 0.260 | 0.250 | 0.240 | 0.230 | 0.330 | 0.240 | 0.000 | 0.250 | 0.290 | 0.240 | 0.240 |
| General-domain Large Vision-Language Models | | | | | | | | | | | | | | | |
| Qwen2-VL-7B | 0.250 | 0.170 | 0.250 | 0.280 | 0.320 | 0.320 | 0.070 | 0.360 | 0.080 | 0.500 | 0.000 | 0.840 | 0.320 | 0.240 | 0.320 |
| Qwen2-VL-72B | 0.240 | 0.180 | 0.270 | 0.330 | 0.350 | 0.240 | 0.060 | 0.410 | 0.070 | 0.360 | 0.000 | 1.000 | 0.000 | 0.290 | 0.320 |
| Ovis1.6-Gemma2-9B | 0.240 | 0.310 | 0.210 | 0.280 | 0.340 | 0.270 | 0.380 | 0.250 | 0.300 | 0.340 | 0.000 | 0.730 | 0.350 | 0.260 | 0.310 |
| InternVL2-8B | 0.260 | 0.310 | 0.250 | 0.240 | 0.300 | 0.250 | 0.480 | 0.340 | 0.120 | 0.490 | 0.000 | 0.790 | 0.310 | 0.270 | 0.270 |
| InternVL2-26B | 0.220 | 0.260 | 0.250 | 0.240 | 0.310 | 0.240 | 0.220 | 0.360 | 0.180 | 0.500 | 0.000 | 0.890 | 0.270 | 0.250 | 0.260 |
| InternVL2-40B | 0.270 | 0.230 | 0.250 | 0.280 | 0.340 | 0.250 | 0.290 | 0.290 | 0.190 | 0.550 | 0.000 | 0.440 | 0.360 | 0.280 | 0.300 |
| LLaVA-1.6-7B | 0.250 | 0.340 | 0.280 | 0.260 | 0.220 | 0.200 | 0.150 | 0.430 | 0.320 | 0.470 | 0.000 | 0.000 | 0.340 | 0.250 | 0.320 |
| LLaVA-1.6-13B | 0.240 | 0.320 | 0.280 | 0.260 | 0.250 | 0.240 | 0.160 | 0.470 | 0.340 | 0.550 | 0.010 | 0.070 | 0.370 | 0.300 | 0.350 |
| Llama3.2-11B | 0.240 | 0.170 | 0.270 | 0.220 | 0.290 | 0.190 | 0.360 | 0.330 | 0.110 | 0.390 | 0.000 | 0.270 | 0.050 | 0.200 | 0.280 |
| GLM-4V-9B | 0.280 | 0.300 | 0.220 | 0.270 | 0.240 | 0.230 | 0.240 | 0.360 | 0.310 | 0.450 | 0.000 | 0.240 | 0.350 | 0.300 | 0.330 |
| DeepSeek-VL-7B | 0.250 | 0.270 | 0.290 | 0.260 | 0.270 | 0.150 | 0.270 | 0.380 | 0.300 | 0.360 | 0.000 | 0.000 | 0.370 | 0.230 | 0.330 |
| MiniCPM-V-2.5 | 0.270 | 0.330 | 0.260 | 0.270 | 0.310 | 0.240 | 0.240 | 0.360 | 0.230 | 0.460 | 0.000 | 0.540 | 0.310 | 0.290 | 0.210 |
| Phi3-Vision | 0.260 | 0.310 | 0.280 | 0.290 | 0.310 | 0.230 | 0.130 | 0.440 | 0.070 | 0.350 | 0.000 | 0.950 | 0.330 | 0.270 | 0.320 |
| mPLUG-Owl3-7B | 0.240 | 0.170 | 0.330 | 0.240 | 0.310 | 0.220 | 0.290 | 0.430 | 0.320 | 0.360 | 0.000 | 0.230 | 0.230 | 0.240 | 0.220 |
| Molmo-7B-D | 0.260 | 0.300 | 0.240 | 0.230 | 0.150 | 0.280 | 0.260 | 0.230 | 0.240 | 0.500 | 0.000 | 0.420 | 0.370 | 0.260 | 0.320 |
| Remote Sensing Large Vision-Language Models | | | | | | | | | | | | | | | |
| GeoChat | 0.250 | 0.260 | 0.220 | 0.250 | 0.260 | 0.220 | 0.270 | 0.500 | 0.220 | 0.390 | 0.030 | 0.090 | 0.260 | 0.250 | 0.340 |
| LHRS-Bot | 0.240 | 0.290 | 0.250 | 0.290 | 0.250 | 0.200 | 0.210 | 0.120 | 0.200 | 0.330 | - | 0.020 | - | 0.260 | 0.310 |
| LHRS-Bot-nova | 0.210 | 0.230 | 0.270 | 0.280 | 0.330 | 0.190 | 0.320 | 0.240 | 0.320 | 0.460 | 0.000 | 0.350 | 0.340 | 0.210 | 0.310 |
| VHM | 0.250 | 0.230 | 0.290 | 0.260 | 0.290 | 0.220 | 0.350 | 0.370 | 0.250 | 0.500 | 0.050 | 0.920 | 0.300 | 0.270 | 0.300 |

Table 12: Fine-grained random and blind evaluation results for Reasoning. Abbreviations adopted: TP for Time Property; PP for Physical Property; EA for Environmental Assessment; RA for Resource Assessment; DD for Disaster Discrimination; GD for Geospatial Determination; SI for Situation Inference.

| Model | Attribute Reasoning | | Assessment Reasoning | | Common Sense Reasoning | | |
|---|---|---|---|---|---|---|---|
| | TP | PP | EA | RA | DD | GD | SI |
| Random | 0.230 | 0.320 | 0.260 | 0.310 | 0.290 | 0.290 | 0.250 |
| General-domain Large Vision-Language Models | | | | | | | |
| Qwen2-VL-7B | 0.250 | 0.230 | 0.220 | 0.130 | 0.240 | 0.260 | 0.240 |
| Qwen2-VL-72B | 0.220 | 0.090 | 0.470 | 0.130 | 0.260 | 0.260 | 0.330 |
| Ovis1.6-Gemma2-9B | 0.270 | 0.360 | 0.420 | 0.260 | 0.260 | 0.250 | 0.280 |
| InternVL2-8B | 0.310 | 0.260 | 0.140 | 0.340 | 0.310 | 0.240 | 0.380 |
| InternVL2-26B | 0.210 | 0.120 | 0.260 | 0.130 | 0.280 | 0.220 | 0.280 |
| InternVL2-40B | 0.220 | 0.260 | 0.150 | 0.050 | 0.240 | 0.300 | 0.330 |
| LLaVA-1.6-7B | 0.250 | 0.330 | 0.140 | 0.360 | 0.250 | 0.260 | 0.270 |
| LLaVA-1.6-13B | 0.230 | 0.400 | 0.170 | 0.300 | 0.230 | 0.260 | 0.320 |
| Llama3.2-11B | 0.220 | 0.210 | 0.390 | 0.100 | 0.250 | 0.250 | 0.350 |
| GLM-4V-9B | 0.270 | 0.340 | 0.210 | 0.390 | 0.220 | 0.200 | 0.240 |
| DeepSeek-VL-7B | 0.220 | 0.370 | 0.130 | 0.340 | 0.250 | 0.270 | 0.370 |
| MiniCPM-V-2.5 | 0.250 | 0.370 | 0.330 | 0.340 | 0.260 | 0.280 | 0.160 |
| Phi3-Vision | 0.200 | 0.180 | 0.160 | 0.180 | 0.240 | 0.250 | 0.160 |
| mPLUG-Owl3-7B | 0.200 | 0.190 | 0.300 | 0.120 | 0.290 | 0.250 | 0.350 |
| Molmo-7B-D | 0.190 | 0.390 | 0.280 | 0.410 | 0.260 | 0.230 | 0.340 |
| Remote Sensing Large Vision-Language Models | | | | | | | |
| GeoChat | 0.200 | 0.420 | 0.270 | 0.380 | 0.220 | 0.250 | 0.250 |
| LHRS-Bot | 0.260 | 0.360 | 0.240 | 0.420 | 0.290 | 0.200 | 0.240 |
| LHRS-Bot-nova | 0.180 | 0.210 | 0.350 | 0.060 | 0.300 | 0.240 | 0.320 |
| VHM | 0.260 | 0.240 | 0.280 | 0.040 | 0.270 | 0.180 | 0.260 |

## E.2 Circular Evaluation

Following the configuration in MMBench [11], we also conducted a circular evaluation by circularly shifting the choices to give a more robust evaluation based on CHOICE. As shown in Figure 14, each question is fed to a VLM $N$ times ($N$ is the number of choices), and VLM is considered successful in answering the question only if it provides the correct answer in all circular passes.

Table 13, 14 and 15 shows the results of circular evaluation. The performance of each model has declined to varying degrees, with RSVLMs showing the most significant drop. Specifically, LHRS-Bot-nova experiences the largest drop in 4 out of 6 L-2 dimensions (ILC, CIR, AttR, and CSR), while LHRS-Bot shows a similar decline in 3 out of 6 L-2 dimensions (ILC, AssR, and CSR). An in-depth analysis of L-3 tasks reveals that several tasks have an accuracy near or even equal to 0,

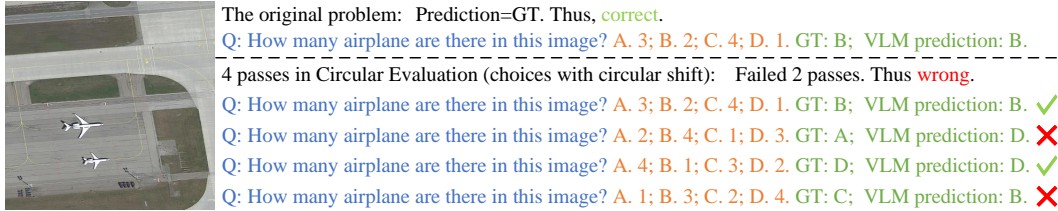

The original problem:  Prediction=GT. Thus, correct.
Q: How many airplane are there in this image? A. 3; B. 2; C. 4; D. 1. GT: B;  VLM prediction: B.
- - - - - - - - - - - - - - - - - - - - - - - - - - - - - - - - - - - - - - - - - - - - - - - - - - - - - - - - - - - - -
4 passes in Circular Evaluation (choices with circular shift):   Failed 2 passes. Thus wrong.
Q: How many airplane are there in this image? A. 3; B. 2; C. 4; D. 1. GT: B;  VLM prediction: B. ✔
Q: How many airplane are there in this image? A. 2; B. 4; C. 1; D. 3. GT: A;  VLM prediction: D. ✘
Q: How many airplane are there in this image? A. 4; B. 1; C. 3; D. 2. GT: D;  VLM prediction: D. ✔
Q: How many airplane are there in this image? A. 1; B. 3; C. 2; D. 4. GT: C;  VLM prediction: B. ✘

Figure 14: Circular Evaluation strategy. A problem is tested multiple times with circularly shifted choices, and the VLM needs to succeed in all testing passes. In this example, the VLM failed in passes 2 and 4, thus considered failed the problem.

Table 13: Circular evaluation results for L-2 dimensions. The negative values in the second column under each L-2 dimension represent the changes in performance after circular evaluation. The largest (second largest) change is in bold red (red). Abbreviations adopted: ILC for Image-level Comprehension; SII for Single-instance Identification; CID for Cross-instance Discernment; AttR for Attribute Reasoning; AssR for Assessment Reasoning; CSR for common sense Reasoning.

| Model | ILC | | SII | | CID | | AttR | | AssR | | CSR | |
|---|---|---|---|---|---|---|---|---|---|---|---|---|
| General-domain Large Vision-Language Models | | | | | | | | | | | | |
| Qwen2-VL-7B | 0.742 | -0.064 | 0.567 | -0.071 | 0.517 | -0.158 | -0.096 | -0.096 | -0.240 | -0.240 | 0.844 | -0.040 |
| Qwen2-VL-72B | 0.809 | -0.045 | 0.645 | -0.056 | 0.636 | -0.106 | -0.076 | -0.076 | -0.158 | -0.158 | 0.867 | -0.047 |
| Ovis1.6-Gemma2-9B | 0.752 | -0.077 | 0.543 | -0.063 | 0.493 | -0.139 | -0.144 | -0.144 | -0.175 | -0.175 | 0.841 | -0.049 |
| InternVL2-8B | 0.701 | -0.088 | 0.483 | -0.083 | 0.490 | -0.158 | -0.164 | -0.164 | -0.225 | -0.225 | 0.746 | -0.070 |
| InternVL2-26B | 0.743 | -0.077 | 0.495 | -0.083 | 0.390 | -0.205 | -0.118 | -0.118 | -0.205 | -0.205 | 0.856 | -0.034 |
| InternVL2-40B | 0.776 | -0.062 | 0.547 | -0.082 | 0.577 | -0.144 | -0.166 | -0.166 | -0.240 | -0.240 | 0.924 | -0.021 |
| LLaVA-1.6-7B | 0.583 | -0.103 | 0.367 | -0.096 | 0.349 | -0.224 | -0.190 | -0.190 | -0.215 | -0.215 | 0.634 | -0.114 |
| LLaVA-1.6-13B | 0.639 | -0.081 | 0.415 | -0.107 | 0.498 | -0.128 | -0.172 | -0.172 | -0.123 | -0.123 | 0.661 | -0.071 |
| Llama3.2-11B | 0.608 | -0.137 | 0.341 | -0.161 | 0.224 | -0.280 | -0.182 | -0.182 | -0.373 | -0.373 | 0.686 | -0.117 |
| GLM-4V-9B | 0.649 | -0.135 | 0.448 | -0.107 | 0.486 | -0.158 | -0.212 | -0.212 | -0.300 | -0.300 | 0.783 | -0.080 |
| DeepSeek-VL-7B | 0.687 | -0.077 | 0.488 | -0.073 | 0.428 | -0.166 | -0.088 | -0.088 | -0.075 | -0.075 | 0.801 | -0.039 |
| MiniCPM-V-2.5 | 0.633 | -0.124 | 0.428 | -0.119 | 0.424 | -0.207 | -0.136 | -0.136 | -0.175 | -0.175 | 0.770 | -0.087 |
| Phi3-Vision | 0.603 | -0.107 | 0.398 | -0.104 | 0.409 | -0.172 | -0.160 | -0.160 | -0.143 | -0.143 | 0.617 | -0.083 |
| mPLUG-Owl3-7B | 0.656 | -0.109 | 0.412 | -0.112 | 0.226 | -0.261 | -0.214 | -0.214 | -0.230 | -0.230 | 0.786 | -0.064 |
| Molmo-7B-D | 0.590 | -0.131 | 0.356 | -0.197 | 0.384 | -0.264 | -0.172 | -0.172 | -0.205 | -0.205 | 0.613 | -0.111 |
| Remote Sensing Large Vision-Language Models | | | | | | | | | | | | |
| GeoChat | 0.501 | -0.142 | 0.365 | -0.104 | 0.251 | -0.229 | -0.202 | -0.202 | -0.274 | -0.274 | 0.570 | -0.127 |
| LHRS-Bot | 0.462 | -0.171 | 0.150 | -0.141 | 0.005 | -0.166 | -0.222 | -0.222 | -0.426 | -0.426 | 0.393 | -0.217 |
| LHRS-Bot-nova | 0.478 | -0.210 | 0.261 | -0.268 | 0.240 | -0.286 | -0.336 | -0.336 | -0.120 | -0.120 | 0.341 | -0.303 |
| VHM | 0.634 | -0.117 | 0.589 | -0.114 | 0.180 | -0.256 | -0.266 | -0.266 | -0.256 | -0.256 | 0.541 | -0.203 |

Table 14: Fine-grained circular evaluation results for Perception. Abbreviations adopted: IM for Image Modality; IQ for Image Quality; MR for Map Recognition; SC for Scene Classification; IC for Image Caption; LR for Landmark Recognitionl; OC for Object Counting; OL for Object Localization; OP for Object Presence; AR for Attribute Recognition; HD for Hallucination Detection; AC for Attribute Comparison; SR for Spatial Relationship; CD for Change Detection.

| Model | Image-Level Comprehension | | | | | | Single-Instance Identification | | | | | Cross-Instance Relationship | | |
|---|---|---|---|---|---|---|---|---|---|---|---|---|---|---|
| | IM | IQ | MR | SC | IC | LR | OC | OL | OP | AR | HD | AC | SR | CD |
| General-domain Large Vision-Language Models | | | | | | | | | | | | | | |
| Qwen2-VL-7B | 0.650 | 0.380 | 0.710 | 0.910 | 0.900 | 0.950 | 0.470 | 0.680 | 0.930 | 0.430 | 0.480 | 0.610 | 0.340 | 0.620 |
| Qwen2-VL-72B | 0.800 | 0.460 | 0.730 | 0.940 | 1.000 | 0.970 | 0.500 | 0.770 | 0.940 | 0.460 | 0.630 | 0.770 | 0.430 | 0.710 |
| Ovis1.6-Gemma2-9B | 0.620 | 0.460 | 0.560 | 0.920 | 0.910 | 0.850 | 0.420 | 0.570 | 0.930 | 0.460 | 0.850 | 0.580 | 0.340 | 0.570 |
| InternVL2-8B | 0.590 | 0.370 | 0.250 | 0.900 | 0.830 | 0.710 | 0.300 | 0.440 | 0.850 | 0.440 | 0.710 | 0.750 | 0.180 | 0.500 |
| InternVL2-26B | 0.650 | 0.400 | 0.480 | 0.930 | 0.820 | 0.970 | 0.320 | 0.510 | 0.940 | 0.440 | 0.700 | 0.530 | 0.120 | 0.550 |
| InternVL2-40B | 0.680 | 0.460 | 0.480 | 0.950 | 0.920 | 0.940 | 0.410 | 0.670 | 0.970 | 0.480 | 0.510 | 0.830 | 0.300 | 0.550 |
| LLaVA-1.6-7B | 0.250 | 0.250 | 0.170 | 0.880 | 0.600 | 0.660 | 0.030 | 0.500 | 0.920 | 0.460 | 0.020 | 0.540 | 0.160 | 0.300 |
| LLaVA-1.6-13B | 0.420 | 0.340 | 0.180 | 0.870 | 0.760 | 0.620 | 0.150 | 0.530 | 0.790 | 0.480 | 0.220 | 0.770 | 0.260 | 0.380 |
| Llama3.2-11B | 0.530 | 0.090 | 0.480 | 0.840 | 0.780 | 0.790 | 0.270 | 0.330 | 0.820 | 0.420 | 0.180 | 0.260 | 0.090 | 0.360 |
| GLM-4V-9B | 0.440 | 0.230 | 0.520 | 0.890 | 0.860 | 0.950 | 0.420 | 0.500 | 0.840 | 0.460 | 0.450 | 0.780 | 0.160 | 0.460 |
| DeepSeek-VL-7B | 0.430 | 0.410 | 0.530 | 0.900 | 0.810 | 0.840 | 0.340 | 0.620 | 0.940 | 0.420 | 0.330 | 0.620 | 0.230 | 0.390 |
| MiniCPM-V-2.5 | 0.450 | 0.160 | 0.500 | 0.880 | 0.900 | 0.750 | 0.290 | 0.360 | 0.880 | 0.480 | 0.510 | 0.650 | 0.130 | 0.470 |
| Phi3-Vision | 0.390 | 0.270 | 0.250 | 0.840 | 0.700 | 0.520 | 0.110 | 0.670 | 0.870 | 0.400 | 0.270 | 0.660 | 0.250 | 0.210 |
| mPLUG-Owl3-7B | 0.600 | 0.220 | 0.480 | 0.850 | 0.830 | 0.880 | 0.320 | 0.520 | 0.910 | 0.370 | 0.250 | 0.210 | 0.110 | 0.430 |
| Molmo-7B-D | 0.420 | 0.350 | 0.140 | 0.770 | 0.780 | 0.540 | 0.160 | 0.400 | 0.820 | 0.450 | 0.320 | 0.650 | 0.090 | 0.360 |
| Remote Sensing Large Vision-Language Models | | | | | | | | | | | | | | |
| GeoChat | 0.118 | 0.068 | 0.255 | 0.857 | 0.395 | 0.710 | 0.012 | 0.644 | 0.928 | 0.256 | 0.000 | 0.463 | 0.073 | 0.145 |
| LHRS-Bot | 0.010 | 0.043 | 0.006 | 0.841 | 0.520 | 0.200 | 0.000 | 0.000 | 0.888 | 0.030 | 0.000 | - | 0.000 | 0.020 |
| LHRS-Bot-nova | 0.084 | 0.021 | 0.006 | 0.849 | 0.550 | 0.420 | 0.000 | 0.118 | 0.926 | 0.154 | 0.066 | 0.446 | 0.000 | 0.240 |
| VHM | 0.505 | 0.215 | 0.013 | 0.917 | 0.475 | 0.560 | 0.002 | 0.752 | 0.914 | 0.376 | 0.898 | 0.054 | 0.260 | 0.280 |

indicating the weak capability in specific remote sensing tasks, such as OC, HD, SR, EA, and RA. This phenomenon suggests that RSVLMs are heavily influenced by different prompts, leading to reduced feasibility. Qwen2-VL-72B and InternVL2-40B continue to secure the top two places as shown in the vanilla results and have a relatively smaller performance drop, further highlighting their

Table 15: Fine-grained circular evaluation results for Reasoning. Abbreviations adopted: TP for Time Property; PP for Physical Property; EA for Environmental Assessment; RA for Resource Assessment; DD for Disaster Discrimination; GD for Geospatial Determination; SI for Situation Inference.

| Model | Attribute Reasoning | | Assessment Reasoning | | Common Sense Reasoning | | |
|---|---|---|---|---|---|---|---|
| | TP | PP | EA | RA | DD | GD | SI |
| **General-domain Large Vision-Language Models** | | | | | | | |
| Qwen2-VL-7B | 0.420 | 0.560 | 0.130 | 0.370 | 0.850 | 0.950 | 0.770 |
| Qwen2-VL-72B | **0.540** | 0.570 | 0.130 | **0.520** | 0.860 | 0.960 | 0.810 |
| Ovis1.6-Gemma2-9B | 0.490 | 0.430 | 0.350 | 0.510 | 0.860 | 0.900 | 0.790 |
| InternVL2-8B | 0.290 | **0.640** | 0.010 | 0.470 | 0.840 | 0.720 | 0.700 |
| InternVL2-26B | 0.440 | 0.500 | **0.530** | 0.400 | 0.830 | 0.950 | 0.810 |
| InternVL2-40B | 0.510 | 0.540 | 0.110 | 0.350 | **0.930** | **0.970** | **0.890** |
| LLaVA-1.6-7B | 0.020 | 0.420 | 0.090 | 0.260 | 0.660 | 0.570 | 0.660 |
| LLaVA-1.6-13B | 0.110 | 0.430 | 0.310 | 0.360 | 0.700 | 0.550 | 0.710 |
| Llama3.2-11B | 0.170 | 0.350 | 0.030 | 0.010 | 0.720 | 0.810 | 0.580 |
| GLM-4V-9B | 0.100 | 0.520 | 0.030 | 0.190 | 0.810 | 0.940 | 0.660 |
| DeepSeek-VL-7B | 0.290 | 0.540 | 0.110 | 0.360 | 0.840 | 0.840 | 0.750 |
| MiniCPM-V-2.5 | 0.140 | 0.370 | 0.430 | 0.280 | 0.880 | 0.810 | 0.670 |
| Phi3-Vision | 0.190 | 0.320 | 0.110 | 0.110 | 0.800 | 0.430 | 0.620 |
| mPLUG-Owl3-7B | 0.070 | 0.300 | 0.260 | 0.060 | 0.880 | 0.760 | 0.740 |
| Molmo-7B-D | 0.340 | 0.380 | 0.500 | 0.290 | 0.890 | 0.340 | 0.610 |
| **Remote Sensing Large Vision-Language Models** | | | | | | | |
| GeoChat | 0.035 | 0.280 | 0.000 | 0.125 | 0.555 | 0.720 | 0.480 |
| LHRS-Bot | 0.000 | 0.240 | 0.000 | 0.000 | 0.390 | 0.135 | 0.567 |
| LHRS-Bot-nova | 0.020 | 0.177 | 0.000 | 0.000 | 0.280 | 0.165 | 0.500 |
| VHM | 0.195 | 0.080 | 0.080 | 0.095 | 0.650 | 0.515 | 0.487 |

strong and robust capabilities in remote sensing tasks. From the perspective of L-2 dimensions, both AttR and AssR have the greatest negative impact on the performance of each VLM, indicating that when faced with problems requiring reasoning based on the perceived visual content, VLMs are significantly less capable and exhibit a high degree of randomness. Therefore, there is an urgent need for further improvement in the reasoning capabilities of VLMs.

## E.3 RES Task

Owing to the intrinsic differences between language and visual modalities, the models evaluated in Section 4 primarily concentrate on text generation tasks and continue to underperform on vision tasks requiring fine-grained output formats, such as segmentation masks. Most existing approaches bridge this gap by incorporating an additional visual decoder [61–64]. Departing from this strategy, [65] introduces a novel text-as-mask method, which adheres to the next-token prediction paradigm of VLMs for easier optimization. To address the lack of evaluation on the Referring Expression Segmentation (RES) task in Section 4, we report the performance of these five representative methods in Table 16. The relatively low mIoU scores highlight the limitations of current general-domain VLMs in handling dense-pixel prediction tasks within remote sensing scenarios. In contrast, RSVLMs specifically trained for this domain demonstrate substantially improved performance in RES tasks, with RemoteSAM [64] achieving particularly notable gains. Furthermore, several qualitative examples for the RES task are illustrated in Figure 15.

Table 16: Evaluation results for RES task.

| Model | Precision | Recall | F1 | mIoU |
|---|---|---|---|---|
| LISA | 0.30 | 0.32 | 0.27 | 0.22 |
| PixelLM | 0.25 | 0.62 | 0.30 | 0.23 |
| Text4Seg | 0.29 | 0.68 | 0.35 | 0.27 |
| Geopix | 0.71 | 0.56 | 0.58 | 0.46 |
| RemoteSAM | **0.84** | **0.70** | **0.72** | **0.64** |

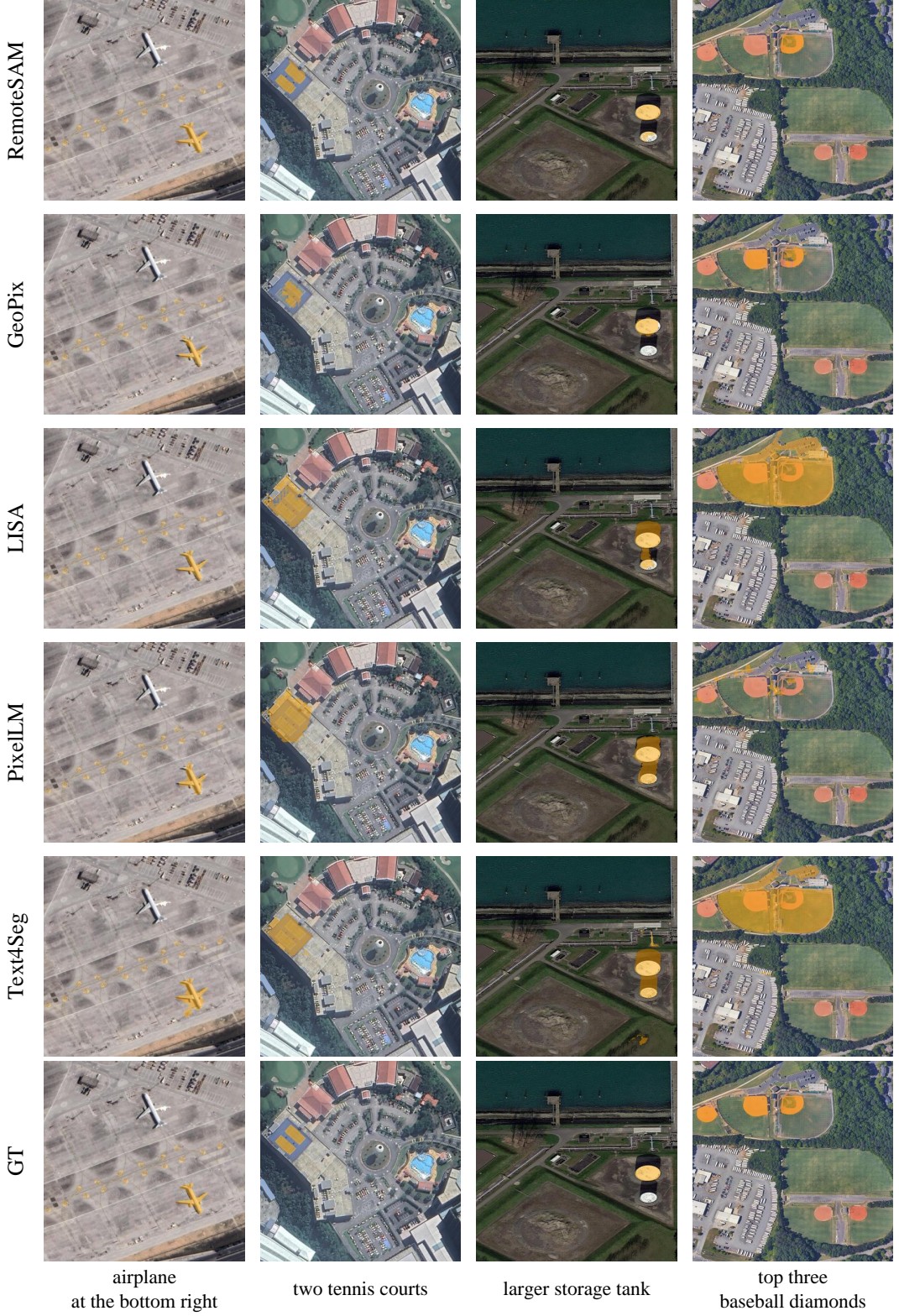

Figure 15: Visualizations on the RES task.

# F Quality Control

## F.1 Quality Control Procedure

We implemented a multi-stage quality control process by recruiting 12 volunteers with professional backgrounds in remote sensing or computer vision, including both master's and doctoral students, to assist in the construction and quality assurance of CHOICE. These volunteers were randomly divided into two groups of six.

During data collection and question creation, two of the six volunteers gather the required RSIs from various data sources, while another two extract and verify the attributes and detailed information of these RSIs. The remaining two take the lead in preliminarily constructing questions based on this information, ensuring that each question aligns accurately with the targeted capabilities.

During the quality inspection process, the six volunteers were further equally divided into two teams, with each team responsible for verifying the quality of specific tasks within CHOICE. A question is approved only if all three team members confirm it is error-free. If one or more volunteers identified an issue, it would be returned to the first group for further discussion and revision.

## F.2 Information of the Human Annotators

We enlisted 12 human annotators with academic backgrounds in remote sensing or computer vision to deeply engage in the quality control stage of the CHOICE dataset. Their responsibilities included collecting suitable RSIs, verifying their attributes, constructing related problems, and ensuring the accuracy and relevance of all tasks and problems in CHOICE. Here is the specific information about these human annotators.

**Education and profession.** Of the 12 human annotators, 6 hold a bachelor's degree, and the other 6 possess a master's degree. In terms of their research areas, 8 are majoring in Photogrammetry and Remote Sensing, while the remaining 4 specialize in Computer Science with a focus on computer vision.

**Age.** Among the 12 human annotators, 6 are aged 21-25, 4 are aged 26-29, and 2 are aged 30-32.

**Ethical Consideration.** All human annotators are from the same research group as the authors and received appropriate compensation for their work. We provided an hourly wage of 50 RMB for each annotator during their part-time assignments, which exceeds the local minimum wage standard. Additionally, we explicitly informed the annotators about the intended use of the data and required them to ensure that the questions included in CHOICE were free from social bias, ethical issues, or privacy concerns throughout the annotation process.

