# OpenReview forum: "CHOICE: Benchmarking the Remote Sensing Capabilities of Large Vision-Language Models"
_NeurIPS.cc/2025/Datasets_and_Benchmarks_Track — NeurIPS 2025 Datasets and Benchmarks Track poster_

### Official Review · Reviewer_Lvaa · 2025-06-12

**Rating:** 6
**Confidence:** 4

**Summary:**

This paper presents CHOICE, a novel benchmark for systematically evaluating VLMs on remote sensing tasks. This benchmark is carefully constructed using data from 50 globally distributed cities, and avoids overlap with publicly available dataset to mitigate evaluation bias from data leakage, a limitation observed in existing benchmarks.

**Dataset Code Accessibility:**

Yes

**Dataset Code Comments:**

The proposed benchmark is publicly available on both HuggingFace and Kaggle, with detailed evaluation codes and a user guide provided on GitHub, ensuring reproducibility.

**Ethical Considerations:**

No, there are no or only very minor ethics concerns

**Final Justification:**

The authors have addressed all my issues, and after considering other reviewers' comments and the authors' responses, I believe that such papers should be encouraged and accepted by the NeurIPS community. Therefore, I raise my socre finally to strong accept and hope a good presentation in Dec!

**Limitations Weaknesses:**

1. I noticed that the benchmark is entirely built from scratch, with all the data newly collected instead of using any publicly available ones. That’s a solid way to ensure a more objective evaluation. I’m still curious how the authors handled data processing and quality control for these new data.
2. Since multiple-choice questions inherently allow models to guess answers based on probabilistic strategies, has the potential inflation of accuracy due to random guessing been considered? Would it be beneficial to include options such as "None of the above" or "Unanswerable" to mitigate this effect?
3. When evaluating visual grounding tasks, are oriented bounding boxes or horizontal bounding boxes used? Clarification is needed, as certain models are not compatible with OBBs.

**Strengths Contributions:**

1. A comprehensive and hierarchical benchmark that covers many RS-specific tasks requiring image-level, instance-level, pixel-level perception and reasoning of satellite imagery to solve.
2. All data is constructed from scratch, rather than derived from existing public datasets, which supports the benchmark’s objectivity and reduces the risk of data leakage.
3. Comprehensive comparisons and experimental analysis across mainstream proprietary and open-source VLMs.

---

> ### Author Rebuttal · Authors · 2025-07-30
>
> We sincerely thank the reviewer for their positive feedback and thoughtful recognition of our contributions. We’re especially grateful for the acknowledgment of our effort invested in constructing the dataset from scratch. Below, we address your specific questions in detail.
>
> > **A1: Further Details on Data Processing and Quality Control**
>
> Thank you for the thoughtful question. During data collection, we leveraged openly accessible satellite data and the OpenStreetMap (OSM) tag system to gather a large pool of candidate remote sensing images (RSIs), such as RGB, SAR, and multi-temporal images. While this gave us broad coverage of the desired categories, many of the collected RSIs initially did not meet our quality or relevance criteria. To address this, we applied multiple traditional vision models (not VLMs) to assess both image-level and instance-level attributes, especially for fine-grained tasks. We retained only high-confidence overlapping outputs across models, focusing on objective and visually determinable attributes such as color, direction, and bounding boxes. This filtering process helped us eliminate most RSIs lacking relevant objects and ensured that retained instances exhibited clear, unambiguous characteristics. Additionally, in both the “Foundation model-driven construction” and “Human-GPT-4 collaboration” stages, model outputs were used only as noisy first-pass references. Human annotators were explicitly instructed not to follow model suggestions blindly. RSIs with false positives were discarded, and those with missed detections were manually corrected. All questions, choices, and labels were verified against the actual RSI content, cross-checked by multiple annotators, and revised whenever model-generated content appeared superficial or biased (see Appendix F for details).
>
> ---
>
> > **A2: Mitigating Guessing Effects in MCQ-Based Evaluation**
>
> Thank you for the valuable suggestion. We have considered the impact of random guessing. Following MMStar[r1], we conducted a “blind” experiment (questions without images) to assess inherent model bias. As shown in Appendix E.1, VLMs perform close to random in this setting, confirming that performance gains in the main benchmark are not due to chance. We also adopted CircularEval from MMBench[r2] for stricter evaluation (Appendix E.2). Moreover, the “hallucination detection” task directly addresses this issue by requiring models to select “The object is not in the picture” when appropriate, effectively testing their ability to reject uncertain or mismatched inputs—similar to your suggestion of adding "none of the above" or "unanswerable" options.
>
> [r1] Chen, Lin, et al. "Are we on the right way for evaluating large vision-language models?." Advances in Neural Information Processing Systems 37 (2024): 27056-27087.
>
> [r2] Liu, Yuan, et al. "Mmbench: Is your multi-modal model an all-around player?." European conference on computer vision. Cham: Springer Nature Switzerland, 2024.
>
> ---
>
> > **A3: Clarification on Bounding Box Format in Visual Grounding Tasks**
>
> We clarify that OBBs are the standard in remote sensing due to arbitrary object orientations from the top-down view and consider the ability to output OBBs as part of the overall capability required for remote sensing applications. Hence, our evaluation uses OBBs. For general-domain VLMs, we compute HBB-to-OBB overlap, while for models like GeoChat and VHM that support OBBs, we compute OBB-to-OBB overlap. Results show that remote sensing VLMs perform better on OBB tasks, highlighting the benefit of domain-specific training and format alignment. We hope that our benchmark encourages broader support for OBB outputs, particularly among general-domain VLMs.

---

> > ### Comment · Reviewer_Lvaa · 2025-08-03
> > **About Final Justification**
> >
> > The authors have addressed all my issues, and after considering other reviewers' comments and the authors' responses, I believe that such papers should be encouraged and accepted by the NeurIPS community. Therefore, I raise my socre finally to strong accept and hope a good presentation in Dec!

---

> > > ### Author Response · Authors · 2025-08-05
> > > **Thank you for your strong support**
> > >
> > > We sincerely thank reviewer Lvaa for the strong support and encouraging feedback! We truly appreciate your recognition of our work and your recommendation to the NeurIPS community. We will continue to improve our benchmark and hope it can make a meaningful contribution to the remote sensing community.

---

### Official Review · Reviewer_KUvG · 2025-06-29

**Rating:** 6
**Confidence:** 4

**Summary:**

A novel benchmark, CHOICE, is introduced to evaluate VLMs in the context of remote sensing. It consists of 23 distinct leaf tasks organized within a three-level hierarchical structure. The authors have carefully collected the data from scratch to avoid potential data leakage. An assessment of 24 VLMs uncovers their primary weaknesses and provides insights for improvements. This “objective” benchmark is expected to be of good interest to the RS community.

**Dataset Code Accessibility:**

Yes

**Ethical Considerations:**

No, there are no or only very minor ethics concerns

**Final Justification:**

The authors addressed all of my concerns, and I believe this work has made a solid contribution to the development of LLM in the remote sensing community. Therefore, I will raise my score to 6.

**Limitations Weaknesses:**

1. While human validation is included, I am concerned that the use of foundation models and GPT-4 in the question construction process may introduce bias. In particular, this approach could favor methods that rely on similar foundation models. Could the authors clarify how such potential bias is mitigated during dataset creation?

2. The majority of the benchmark consists of RGB images, with other data modalities, such as SAR and multi-spectral data, being limited. The authors are encouraged to include a broader range of RS-specific modalities for future improvements.

**Strengths Contributions:**

1. A novel benchmark for the domain of remote sensing, featuring interesting and context-specific reasoning tasks related to environmental factors.

2. The dataset excludes public datasets to ensure “objectivity”. A detailed data construction process is implemented, along with quality control that involves human verification.

3. This paper is well-written. The benchmark is publicly available and the evaluation codes are transparent and reproducible.

---

> ### Author Rebuttal · Authors · 2025-07-30
>
> We sincerely thank the reviewer for the positive feedback and thoughtful recognition of our work. We are glad that the novelty of our benchmark, the domain-specific reasoning tasks, and the efforts in data construction and quality control were well received. Below, we address your specific questions in detail.
>
> > **A1: Mitigation of Potential Bias in Dataset Construction**
>
> We agree that using GPT-4 in the construction of Image Caption and Change Detection tasks may in principle introduce subtle biases, which could stem from the training data of GPT-4V. However, as GPT-4V is a closed model, directly assessing such bias is challenging without a dedicated human study. To mitigate potential bias, we emphasize the following:
> 1. In both the “Foundation model-driven construction” and “Human-GPT-4 collaboration” stages, model outputs were treated as noisy first-pass references only. Human annotators were explicitly instructed not to follow model suggestions blindly. All questions, choices, and labels were verified against the RSI content, cross-checked by multiple annotators, and revised when model-generated content appeared superficial or biased (see Appendix F).
> 2. For “Foundation model-driven” tasks, we relied on multiple traditional visual models (not VLMs), selecting only high-confidence overlapping outputs—focused on determinant objective attributes like color, direction, and bounding boxes. Biased or incorrect outputs were manually filtered or corrected.
> 3. If the GPT-4-generated data favored GPT-like architectures, we would expect those models to consistently outperform others. However, as shown in Tables 2 and 4, open-source models such as Qwen2-VL-72B and VHM perform on par with or better than GPT-4o on key tasks, suggesting no systematic advantage was introduced.
>
> ---
>
> > **A2: Inclusion of a Broader Range of Remote Sensing Modalities in Future Work**
>
> We acknowledge that the majority of our dataset consists of RGB images, with other remote sensing modalities, such as SAR, multi-spectral, and nighttime light images, being limited. These modalities are included only in the Image Modality task, which specifically aims to assess the ability of VLMs to recognize and differentiate between different data modalities. The task is not directly related to content interpretation. We are aware of the value of incorporating a broader range of remote sensing modalities and are actively planning to include more diverse sources of remote sensing imagery in future iterations of this work. This is further discussed in the "Conclusion and Limitations" section.

---

> > ### Comment · Reviewer_KUvG · 2025-08-04
> > **Decision**
> >
> > The authors addressed all of my concerns, and I believe this work has made a solid contribution to the development of LLM in the remote sensing community. Therefore, I will raise my score to 6.

---

> > > ### Author Response · Authors · 2025-08-05
> > > **Thank you for the encouraging feedback**
> > >
> > > We thank reviewer KUvG for the encouraging feedback and for recognizing our efforts! We hope that our proposed benchmark can serve as a valuable resource for the remote sensing community. We are committed to continuously improving it and exploring further applications in this field.

---

### Official Review · Reviewer_99YZ · 2025-07-01

**Rating:** 4
**Confidence:** 4

**Summary:**

The CHOICE benchmark presents a valuable resource to evaluate the abilities of Large Language Models (LLMs) on the remote sensing data. The benchmark is designed with three levels of tasks across multiple RS domains. It draws from multiple data and label sources and aims to assess the capability of LLMs in both the general and the RS domains. The authors evaluate several LLMs, including commercial models and open-source ones, highlighting strengths and limitations in their remote sensing understanding and reasoning. The findings are interesting and useful.

**Additional Feedback:**

Please consider incorporating metadata or hybrid (visual & text) inputs to evaluate multi-modal LLM capabilities.

**Dataset Code Accessibility:**

Yes

**Dataset Code Comments:**

The code is complete and well-documented.

**Ethical Comments:**

No ethics concerns

**Ethical Considerations:**

No, there are no or only very minor ethics concerns

**Final Justification:**

The authors addressed some of my concerns. However, I still think that there is room for improvement regarding the dataset and experiment design. Thus, I choose to keep my rating to a Borderline accept.

**Limitations Weaknesses:**

1. One major limitation is the lack of awareness or contextual understanding of geographic specificity. In practical RS applications, models should make use of location-specific knowledge. The benchmark currently does not account for this, which lacks real-world RS tasks that require deeper geospatial reasoning.

2. While the inclusion of a reasoning category is good, the current tasks remain relatively shallow. They often involve simple deductive steps or domain recall, rather than requiring multi-step, cross-modal, or decision-based reasoning typical in applied RS tasks like disaster response, land-use change detection, or multi-temporal analysis.

3. The benchmark lacks time-series data and metadata of RS images. Future iterations could benefit from integrating spatial-temporal inference or hybrid tasks that combine imagery with RS metadata.

**Strengths Contributions:**

1. The emergence of LLMs has transformed AI across domains, but RS has remained underrepresented in LLM evaluation. CHOICE addresses this gap with a well-structured benchmark tailored to RS challenges.

2. The construction pipeline is logical and well-motivated, including careful selection of data sources, quality control, and label curation. The three-level task framework adds useful granularity for evaluating LLM capabilities.

3. The benchmarking results reveal the current limitations of LLMs in handling domain-specific RS knowledge, especially when it comes to complex reasoning. These insights are valuable for guiding future LLM development tailored to RS domains.

---

> ### Author Rebuttal · Authors · 2025-07-30
>
> We’re pleased that you found CHOICE to be a well-structured benchmark that addresses the underexplored area of VLM evaluation. Your observations on the current limitations of VLMs in domain-specific reasoning closely align with our motivations. Below, we provide detailed responses to your specific questions.
>
> > **A1: Geographic Specificity and Contextual Geospatial Reasoning**
>
> We sincerely thank the reviewer for highlighting this important limitation. We fully agree that geographic specificity and contextual geospatial reasoning are essential components of real-world remote sensing (RS) applications, and we appreciate the opportunity to further clarify the current scope of CHOICE and our plans for future enhancement. Although CHOICE currently does not include tasks explicitly designed to assess location-specific contextual knowledge, it does incorporate several elements that touch upon this direction. In particular, tasks such as Geospatial Determination (GD) and Landmark Recognition (LR) require models to associate visual content with geographic or cultural semantics, which implicitly tests some basic level of geographic awareness. Furthermore, the global coverage of our dataset—spanning 50 cities across diverse continents and landforms—was intentionally designed to encourage generalization across geographically varied environments, which lays the foundation for more region-aware evaluation.  In future iterations of CHOICE, we plan to extend the benchmark to include tasks such as location-aware semantic reasoning, context-dependent classification, and geo-tagged temporal inference, which will more directly evaluate a model’s ability to utilize geographic priors and contextual cues for spatially grounded understanding.
>
> ---
>
> > **A2: Response to the Concern on Depth of Reasoning Tasks**
>
> We thank the reviewer for highlighting the importance of incorporating deeper and more complex reasoning tasks in remote sensing evaluation. We agree that multi-step, decision-based, and cross-modal reasoning is essential for more thoroughly assessing the practical utility of VLMs. In CHOICE, we adopted a design strategy that prioritizes broad hierarchical coverage (23 leaf tasks across 6 dimensions) and objective, scalable evaluation. This necessarily involves a trade-off, limiting the inclusion of open-ended, multi-step reasoning tasks. However, CHOICE does include several tasks that go beyond shallow recall or direct observation, such as:
> - Change Detection (CD): analyzing multi-temporal imagery to identify structural or land-cover changes;
> - Assessment Reasoning (RA/EA): estimating population density or CO₂ emissions based on geospatial context;
> - Disaster Discrimination and Situation Inference (DD/SI): applying contextual and commonsense reasoning to infer real-world scenarios.
>
> As also shown in our experimental results, most VLMs, including some proprietary and domain-specific models, still perform relatively poorly on these reasoning tasks, highlighting the current limitations of models in handling even moderately complex RS inference problems. We are committed to maintaining and regularly updating the dataset to incorporate more data and in-depth tasks, ensuring its long-term value to the community. Looking ahead, we plan to significantly expand the reasoning dimension by incorporating tasks with the following features:
>
> - Multi-step and sequential reasoning, especially for dynamic scenarios;
> - Longer-term temporal reasoning across sequences of RSIs;
> - Cross-modal integration, such as combining image, metadata (e.g., timestamps, geo-tags), and auxiliary layers (e.g., SAR, multispectral) to emulate real RS workflows. For example, we have constructed interpretation and reasoning tasks based on pre-disaster optical and post-disaster SAR imagery, as summarized in Table R1 below.
> - Decision-making under uncertainty, which aligns more closely with operational use cases like disaster response and urban planning.
>
> Table R1. Performances of general-domain VLMs and RSVLMs on disaster-related tasks. BDC: Building Damage Counting; BDL: Building Damage Localization; BDA: Building Damage Assessment; FIA: Flood Impact Assessment.
> | **Model**        | **BDC** | **BDL** | **BDA** | **FIA** |
> |------------------|--------:|--------:|--------:|--------:|
> | GPT-4o-mini      |   0.26  |   0.33  |   0.79  |   0.32  |
> | Qwen2-VL-7B      |   0.26  |   **0.64**  |   **0.92**  |   0.15  |
> | InternVL2-8B     |   **0.30**  |   0.10  |   0.85  |   0.17  |
> | TeoChat          |   0.26  |   0.38  |   0.61  |   **0.36**  |
>
> ---
>
> > **A3: Enhancing CHOICE with Time-Series and Metadata-Informed Tasks**
>
> We sincerely thank the reviewer for the thoughtful and constructive suggestion. While CHOICE currently includes both bi-temporal and multi-temporal tasks, such as change detection and seasonal inference, that partially explore temporal reasoning, we acknowledge that it does not yet support full-fledged time-series data with continuous temporal granularity. Similarly, although some metadata-related information (e.g., modality, resolution, seasonality, geolocation) is implicitly embedded within certain tasks, we agree that explicitly integrating structured metadata (such as latitude and longitude) could enable richer forms of reasoning and significantly broaden the benchmark’s applicability. In future iterations of CHOICE, we plan to incorporate structured multi-temporal RS data and design new hybrid tasks that involve metadata-informed interpretation.

---

### Official Review · Reviewer_gSdc · 2025-07-19

**Rating:** 5
**Confidence:** 5

**Summary:**

This paper introduces a meticulously structured benchmark for assessing vision language models in remote sensing. CHOICE assesses two main capabilities—perception and reasoning—through 6 secondary dimensions and 23 specialized tasks across 10,507 carefully curated problems. Evaluations of 24 VLMs highlight key model limitations and emphasize the potential of open-source VLMs.

**Dataset Code Accessibility:**

Yes

**Dataset Code Comments:**

CHOICE is publicly available on HuggingFace, and the reproducible evaluation codes are also accessible in GitHub.

**Ethical Considerations:**

No, there are no or only very minor ethics concerns

**Final Justification:**

The author's response is quite convincing to me, and I think this article is very interesting.

**Limitations Weaknesses:**

1.	As shown in Table 5 (Appendix), the Scene Classification task comprises 2k samples, while others have a few hundred. Why did the authors choose such a sample imbalance?
2.	Although three general domain VLMs with pixel level capability were tested, the evaluation should also include RS-specific pixel level segmentation models (e.g., GeoPix[R1], RemoteSAM[R2]) to thoroughly assess fine grained segmentation performance.
[R1] Ou, Ruizhe, et al. "GeoPix: A multimodal large language model for pixel-level image understanding in remote sensing." IEEE Geoscience and Remote Sensing Magazine (2025).
[R2] Yao, Liang, et al. "RemoteSAM: Towards Segment Anything for Earth Observation." arXiv preprint arXiv:2505.18022 (2025).

**Strengths Contributions:**

1.	Detailed hierarchical evaluation structure clearly identifies model strengths and weaknesses.
2.	A rigorous methodology for data collection and quality assurance is employed, effectively avoiding common pitfalls such as data leakage.
3.	Demonstrates that open-source models can be viable alternatives or even superior to proprietary models in specialized tasks, which has strong practical implications for resource-constrained applications.

---

> ### Author Rebuttal · Authors · 2025-07-30
>
> Thank you for recognizing the key strengths of our work. We sincerely appreciate your acknowledgment of our detailed hierarchical evaluation framework, our rigorous data collection and quality assurance methodology. Below, we provide responses to your specific questions.
>
> > **A1: Rationale for Sample Size in Scene Classification Task**
>
> While the Scene Classification task may seem relatively straightforward, it involves multiple categories, with a total of 20 distinct classes in our benchmark. To ensure a fair and balanced evaluation, we opted for a larger sample size of 2,000 for this task. This decision was made to prevent any potential bias toward specific categories, and to guarantee that each category has a sufficient number of samples (approximately 100 per category). By doing so, we aim to provide a more reliable and robust assessment of the model’s performance across all categories, ensuring the results reflect a comprehensive understanding of the scene classification task as a whole.
>
> ---
>
> > **A2: Inclusion of RS-Specific Pixel-Level Segmentation Models**
>
> Thank you for the valuable suggestion. In response, we have conducted additional experiments to include RS-specific VLMs with pixel-level capability, such as GeoPix and RemoteSAM, on our Referring Expression Segmentation (RES) task. Furthermore, since both models are also capable of performing the Visual Grounding (VG) task, we have included corresponding experiments to assess their performance. The updated results, presented in the tables below, offer a more comprehensive comparison and provide a detailed evaluation of fine-grained RES and VG performance.
>
> Table R1. Referring Expression Segmentation performances of general-domain VLMs and RSVLMs.
> | **Model**    | **Precision** | **Recall** | **F1** | **mIoU** |
> |--------------|---------------|------------|--------|----------|
> | LISA         | 0.30          | 0.32       | 0.27   | 0.22     |
> | PixelLM      | 0.25          | 0.62       | 0.30   | 0.23     |
> | Text4Seg     | 0.29          | 0.68       | 0.35   | 0.27     |
> | Geopix       | 0.71          | 0.56       | 0.58   | 0.46     |
> | RemoteSAM    | **0.84**      | **0.70**   | **0.72** | **0.64** |
>
> Table R2 . Visual Grounding performances of GeoPix and RemoteSAM.
> | **Model**    | **Acc\@0.5** |
> |--------------|---------------|
> | GeoPix       | 0.3817          |
> | RemoteSAM    | **0.6283**      |

---

> > ### Comment · Reviewer_gSdc · 2025-08-05
> >
> > According to the author's rebuttal, the author's response is consistent with my understanding, and I believe this article can be accepted.

---

> > > ### Author Response · Authors · 2025-08-05
> > > **Thank you for your support**
> > >
> > > We thank the reviewer gSdc for the positive feedback and for agreeing with our clarifications. We appreciate your support and will continue improving our work to better serve the RS community.

---

### Note · Authors · 2025-08-12

We would like to express our sincere gratitude to the reviewers for their thoughtful and thorough evaluations of our paper. We deeply appreciate the time and effort they have dedicated to reviewing our work. The reviewers have highlighted several strengths of our paper, including:

- The emergence of VLMs has transformed AI across domains, but RS has remained underrepresented in VLM evaluation. Our CHOICE benchmark addresses this gap by providing a comprehensive, well-structured framework specifically designed for RS tasks (`99YZ`).

- Multiple reviewers recognized the detailed hierarchical evaluation framework, which clearly identifies the strengths and weaknesses of the model. They praised the logical construction pipeline, including careful data selection and quality assurance processes to prevent common issues such as data leakage (`gSdc`, `KUvG`, `Lvaa`).

- Reviewers noted the careful construction of all data from scratch, ensuring no overlap with publicly available datasets. This approach ensures objectivity and minimizes the risk of data leakage (`Lvaa`, `KUvG`). Moreover, the transparency and reproducibility of the evaluation code were recognized as essential strengths of the paper (`KUvG`).

- The paper’s demonstration that open-source models can be viable alternatives—if not superior—to proprietary models in specialized tasks was appreciated, particularly for its strong practical implications in resource-constrained applications (`gSdc`).

- The reviewers acknowledged that the benchmarking results offer valuable insights into the limitations of current models, especially in the context of domain-specific remote sensing knowledge and reasoning tasks (`99YZ`, `gSdc`).

We are truly grateful for the reviewers’ constructive feedback and thoughtful evaluations. From their responses, we learned that Reviewer `Lvaa` and `KUvG` raised their score to the highest level. Additionally, we appreciate the positive feedback and high ratings from Reviewer `gSdc` and `99YZ`. Their support of the paper’s contributions has been invaluable, and we greatly appreciate the time and effort they have invested in reviewing our work. We are also committed to continuing to refine and improve this benchmark, aiming to make a more valuable contribution to the remote sensing community.

---

### Decision · Program_Chairs · 2025-09-18

**Decision:**

Accept (poster)

**Comment:**

AC and reviewers thank the authors for the rebuttal. After rebuttal, reviewers converged toward acceptance (two strong accept, one accept, one borderline accept). Overall the paper introduces a well-scoped and carefully curated RS-VLM benchmark with broad, reproducible baselines and leakage-mitigation; concerns about MCQ guessing, limited modality/temporal coverage, and potential GPT-4 bias were reasonably addressed during the rebuttal stage. Considering all these factors, the AC would like to accept the paper. Congrats! Authors are encouraged to incorporate the reviewer-requested additions and clarifications from the discussion to further strengthen the paper.